EMBO
Molecular Medicine

# An intranasally delivered peptide drug ameliorates cognitive decline in Alzheimer transgenic mice

Yu-Sung Cheng[1,†], Zih-ten Chen[2,†], Tai-Yan Liao[2], Chen Lin[2] iD, Howard C-H Shen[2], Ya-Han Wang[2,3], Chi-Wei Chang[4], Ren-Shyan Liu[4,5], Rita P-Y Chen[2,3,*] iD & Pang-hsien Tu[1,**] iD

## Abstract

**Alzheimer's disease (AD) is the most common neurodegenerative disease. Imbalance between the production and clearance of amyloid β (Aβ) peptides is considered to be the primary mechanism of AD pathogenesis. This amyloid hypothesis is supported by the recent success of the human anti-amyloid antibody aducanumab, in clearing plaque and slowing clinical impairment in prodromal or mild patients in a phase Ib trial. Here, a peptide combining polyarginines (polyR) (for charge repulsion) and a segment derived from the core region of Aβ amyloid (for sequence recognition) was designed. The efficacy of the designed peptide, R$_8$-Aβ(25–35), on amyloid reduction and the improvement of cognitive functions were evaluated using _APP/PS1_ double transgenic mice. Daily intranasal administration of PEI-conjugated R$_8$-Aβ(25–35) peptide significantly reduced Aβ amyloid accumulation and ameliorated the memory deficits of the transgenic mice. Intranasal administration is a feasible route for peptide delivery. The modular design combining polyR and aggregate-forming segments produced a desirable therapeutic effect and could be easily adopted to design therapeutic peptides for other proteinaceous aggregate-associated diseases.**

**Keywords** Aβ; Alzheimer disease; peptide therapy; polyarginine; polyethylenimine

**Subject Categories** Neuroscience; Pharmacology & Drug Discovery

## Introduction

Alzheimer's disease (AD) is the most common neurodegenerative disease that causes dementia across multiple cognitive domains. Its incidence increases significantly with age and doubles every 5 years among the geriatric population ≥ 65 years of age. Despite remarkable scientific advancement and the vast resources invested in drug development, no effective therapy is currently available for AD. Thus, it is listed as one of the major unmet medical needs worldwide.

Although the etiology of AD remains unclear, the amyloid cascade hypothesis is the most supported explanation to date and is recently further strengthened by the finding of a protective APP mutation near the β-cleavage site against the development of late-onset dementia (Jonsson _et al_, 2012). Unfortunately, a series of clinical trials based on amyloid reduction therapy (ART) failed to deliver anticipated clinical improvement on mild-to-moderate patients with AD (Ross & Imbimbo, 2010; Aisen _et al_, 2011; Roher _et al_, 2011; Grundman _et al_, 2013; Khorassani & Hilas, 2013), raising legitimate concerns for the accuracy of amyloid cascade hypothesis and the future of ART (Extance, 2010). However, given that alternative strategies aimed at reducing neuroinflammation, cholesterol level, or oxidative stress have similarly failed to improve the clinical outcome of AD, it is fair to say that the jury is still out on finding the culprit for the failures of these clinical trials. Previous data reveal that a substantial percentage (~50%) of neurons have already disappeared in even the mild cognitive impairment or very mild AD (Gomez-Isla _et al_, 1996; Mufson _et al_, 2000; Price _et al_, 2001). Consistent with these, more recent findings show that alterations in amyloid biology represent the earliest detectable changes in the brain in familial AD and start in the brain more than 20 years prior to the onset of AD (Bateman _et al_, 2012). It is likely that the ART trials (and other alternatives) fail because they miss the most opportune "therapeutic window" of AD. Thus, ART still remains a vital and important choice when given earlier. In fact, clinical trials with very early or pre-symptomatic intervention using ART are currently being conducted or have been planned (Miller, 2012; Wadman, 2012; Moulder _et al_, 2013). The preliminary success of ART with aducanumab immunotherapy in decreasing cognitive decline further strengthens this hypothesis (Moreth _et al_, 2013; Lannfelt _et al_, 2014; Ratner, 2015; Underwood, 2015; Selkoe & Hardy, 2016; Sevigny _et al_, 2016).

1 Institute of Biomedical Sciences, Academia Sinica, Taipei, Taiwan
2 Institute of Biological Chemistry, Academia Sinica, Taipei, Taiwan
3 Institute of Biochemical Sciences, National Taiwan University, Taipei, Taiwan
4 Biomedical Imaging Research Center, Department of Nuclear Medicine, National Yang-Ming University and Taipei Veterans General Hospital, Taipei, Taiwan
5 Molecular and Genetic Imaging Core, Taiwan Mouse Clinic, Academia Sinica, Taipei, Taiwan
*Corresponding author. Tel: +886 2 27855696; Fax: +886 2 27889759; E-mail: pyc@gate.sinica.edu.tw
**Corresponding author. Tel: +886 2 26523532; Fax: +886 2 27827654; E-mail: benrosetu@yahoo.com
† These authors contributed equally to this work

Peptide drugs have been used with consistent benefits for many years and have advantages over small molecules, such as higher potency and fewer off-target side effects (Craik *et al*, 2013). In addition, the properties of easy customization and synthesis under a well-controlled environment make peptides excellent candidates for AD drug development. Neurodegenerative diseases encompass a heterogeneous group of neurological diseases characterized by synoptic and neuronal losses caused by multiple factors. Misfolded proteinaceous aggregates which exist in a variety of these diseases besides AD, including Parkinson's disease, Huntington's disease, amyotrophic lateral sclerosis, are considered one of them, and may cause or contribute to these diseases through their prionlike property (Kim & Holtzman, 2010; de Calignon *et al*, 2012; Luk *et al*, 2012; Smethurst *et al*, 2016). In spite of the difference in the constituent proteins and complexity of assembly mechanism, the proteinaceous aggregates across these diseases share common structural conformations such as a β-sheet conformation in the backbone (Funke & Willbold, 2012). This provides the basis for a rational design of therapeutic peptides for these misfolded aggregate-associated diseases by applying a universal principle to reverse the process of formation. In this study, we propose a novel modular approach to design an ART peptide drug and test its efficacy using the *APP/PS1* transgenic mouse model.

# Results

### Design of the inhibitor peptide

Amyloid in AD is an insoluble β-sheet structure formed by Aβ peptides. Thus, in order to inhibit amyloid propagation, the inhibitor should be equipped with the following characteristics: (i) the ability to interact with Aβ monomer/aggregates and (ii) the ability to prevent Aβ from association into higher order polymers and fibrils. Here, we proposed a rational approach based on the concept of modularity to design bipartite inhibitor peptides comprising two different modules, each possessing one of the aforementioned characteristics, as a prototype of potential therapeutic agents. The concept of our design was schematically presented in Fig 1A. The first module was a partial sequence derived from Aβ peptide which, because of its known propensity to self-aggregate, was anticipated to bestow on the inhibitor peptide an ability to bind to Aβ with high specificity. The second module was a charged moiety which, through the repulsion force exerted by these charges, could prevent not only self-aggregation of the inhibitor peptide, but also the propagation of amyloid after the inhibitor peptide bound to its Aβ target.

It has been reported that the sequence of residues 25–35 of Aβ is important for Aβ aggregation and toxicity (Hughes *et al*, 2000; Ban *et al*, 2004; Liu *et al*, 2004). In this study, we designed an L-form inhibitor peptide, $R_8$-Aβ(25–35), and its D-form counterpart, $^DR_8$-Aβ(25–35), by combining Aβ(25–35) (the Aβ-binding module) with a segment of consecutive eight L-form or D-form Arg residues ($R_8$ or $^DR_8$) (the charge repulsion module). The conformational properties of these two bipartite peptides and their effects on the inhibition of Aβ amyloidogenesis and toxicity were examined. Intranasal delivery has been shown to be an effective way to deliver insulin into brain to alleviate memory deficit in patients with amnestic mild cognitive

impairment or AD (Craft *et al*, 2012; Claxton *et al*, 2015). It has been reported that PEI cationization facilitates protein transduction across the cell membrane and has been used in drug design to bypass blood–brain barrier into the brain parenchyma via intranasal delivery (Futami *et al*, 2005, 2007; Kitazoe *et al*, 2005, 2010; Loftus *et al*, 2006; Murata *et al*, 2008a,b; Lin *et al*, 2016). Thus, to facilitate entry of our peptide into the mouse brain, polyethylenimine (PEI) was conjugated with $R_8$-Aβ(25–35), which contained the poly-R segment that could also enhance peptide penetration (Herve *et al*, 2008; Patel *et al*, 2009). We synthesized PEI-conjugated $R_8$-Aβ(25–35) and tested its therapeutic effect in the APP/PS1 transgenic mice via intranasal administration.

### Conformational study of Aβ$_{40}$, $R_8$-Aβ(25–35), and $^DR_8$-Aβ(25–35)

To investigate the biophysical property of our designed peptides concerning amyloid formation, Aβ$_{40}$, $R_8$-Aβ(25–35), and $^DR_8$-Aβ(25–35) dissolved in 20 mM sodium phosphate buffer with 150 mM KCl (pH 7) were individually incubated at 25°C. The circular dichroism (CD) spectra were recorded at various time points as indicated. In Aβ$_{40}$ spectra, as expected, the intensity of the negative ellipticity at 218 nm increased and the intensity at 200 nm decreased with time (Fig 1B and Appendix Fig S1A), consistent with its known ability to form amyloid fibrils as shown by transmission electron microscopy (TEM) (Fig 1F and Appendix Fig S1D). In contrast, the CD spectra of $R_8$-Aβ(25–35) showed negative ellipticity at 200 nm, indicative of random coil structure (Appendix Fig S1B). Similarly, the $^DR_8$-Aβ(25–35) also had CD spectra consistent with random coil structure, which lacked strong negative ellipticity at 200 nm due to the presence of eight D-form arginines in the peptide (Appendix Fig S1C). The CD spectra of both peptides remained largely unchanged with the incubation time. Under the same incubation condition, $R_8$-Aβ(25–35) and $^DR_8$-Aβ(25–35) did not form amyloid fibrils at all except for small blobs of amorphous aggregates under transmission electron microscopy after incubation for 168 h (Appendix Fig S1E and F).

### Inhibition effect of our designed bipartite peptides on the fibrillization of Aβ$_{40}$

To examine whether $R_8$-Aβ(25–35) could interfere with the amyloidogenesis of Aβ$_{40}$, the CD spectra of Aβ$_{40}$ mixed with $R_8$-Aβ(25–35) at 1:0.1, 1:0.2, or 1:1 molar ratios were measured. As shown in Fig 1C–E, the change in the CD spectrum of Aβ$_{40}$ (an increase in the negative ellipticity at 218 nm and a decrease at 200 nm) clearly decreased by $R_8$-Aβ(25–35) in a dose-dependent manner. Notably, the inhibition effect of $R_8$-Aβ(25–35) on Aβ$_{40}$ fibrillization was observed even when its concentration was ten times lower than Aβ$_{40}$ (Fig 1C and E). Consistent with these CD studies, TEM revealed a clear reduction in both thickness and abundance of the amyloid fibrils at all time points we observed (Fig 1G–I). These results showed that $R_8$-Aβ(25–35) interacted with Aβ$_{40}$, interfered with its self-aggregation, and thereby significantly delayed or decreased the formation of Aβ$_{40}$ amyloid fibrils.

Interestingly, the inhibition effect was also observed with $^DR_8$-Aβ(25–35) (Appendix Fig S2), suggesting that it was the charge, rather than the steric structure, of the $R_8$ moiety that prevented Aβ$_{40}$ from aggregation.

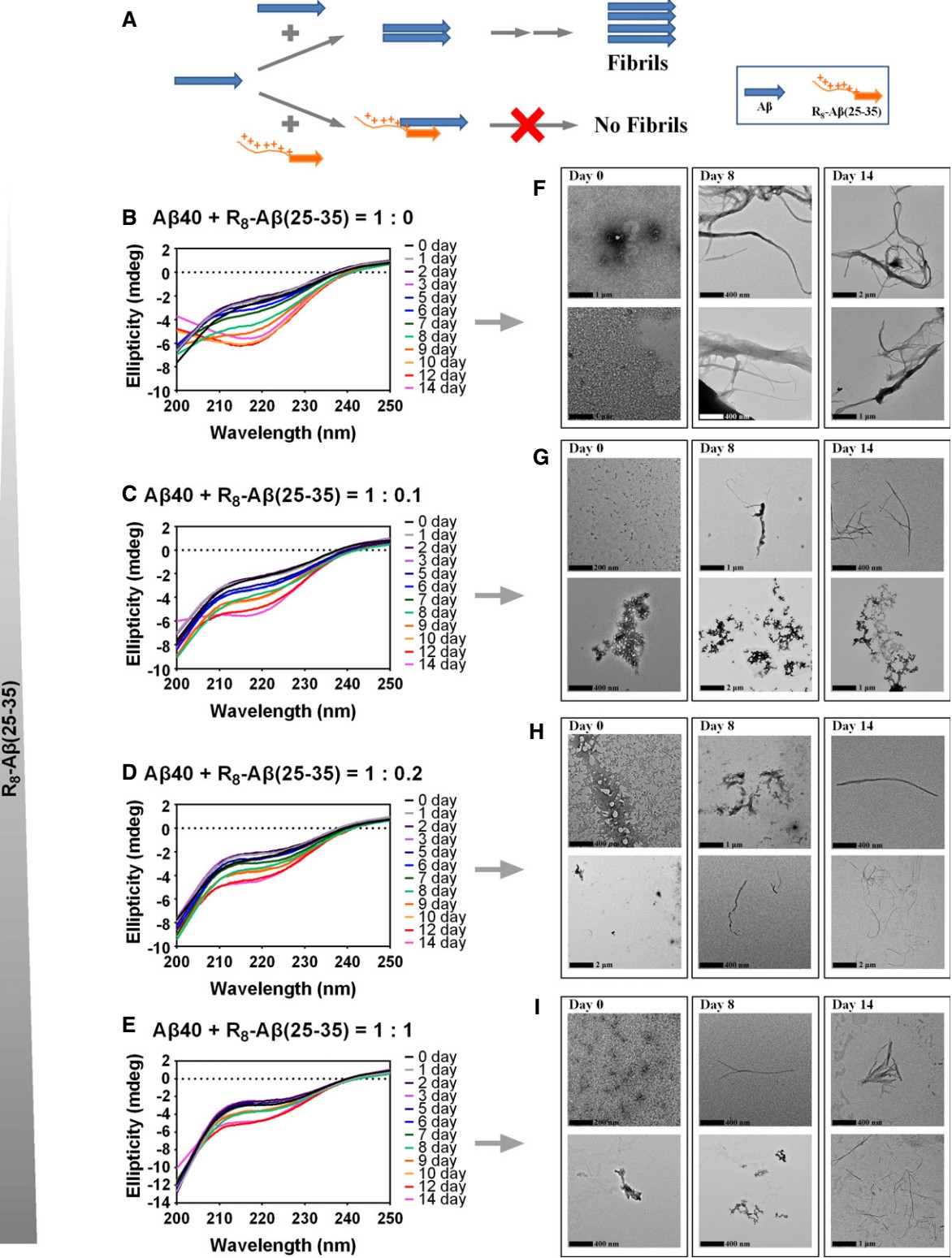

**Figure 1.  The inhibition model of $R_8$-A$\beta$(25–35) and the dose-dependent effect of $R_8$-A$\beta$(25–35) on inhibition of A$\beta_{40}$ fibrillization.**

A$\beta_{40}$ was mixed with $R_8$-A$\beta$(25–35) in different mixing ratios (A$\beta_{40}$:$R_8$-A$\beta$(25–35) = 1:0.1, 1:0.2, 1:1). The A$\beta_{40}$ concentration is 30 μM. The peptides were dissolved in 20 mM sodium phosphate buffer with 150 mM KCl (pH 7) and incubated at 25°C.

A      Proposed working mechanism.

B–E  CD spectra of A$\beta_{40}$ alone (B) and three A$\beta_{40}$/$R_8$-A$\beta$(25–35) mixtures (C, 1:0.1; D, 1:0.2; E, 1:1) were recorded at the indicated incubation times.

F–I  TEM images of the samples in (B), (C), (D), and (E) taken at the indicated incubation times are shown in (F), (G), (H), and (I), respectively.

### Attenuation of $A\beta_{40}$ cytotoxicity by $R_8$-A$\beta$(25–35) and $^DR_8$-A$\beta$(25–35)

Because our designed bipartite peptides interfered with $A\beta_{40}$ self-aggregation, we tested whether these peptides could decrease the toxicity of $A\beta_{40}$ by measuring the viability of Neuro2a, a mouse neuroblastoma cell line with the MTT assay. Cells were treated with peptides as indicated. $A\beta_{40}$ (30 μM) exerted significant cytotoxicity to Neuro2a cells; only 30% of the cells survived the treatment. In contrast, $R_8$-A$\beta$(25–35) or $^DR_8$-A$\beta$(25–35) had no detectable toxicity to the N2a cells (Fig 2A). Interestingly, $R_8$-A$\beta$(25–35) or $^DR_8$-A$\beta$(25–35) each decreased $A\beta_{40}$ toxicity, as evidenced by an increase in cell viability from 30% to 70–75% (Fig 2B). For comparison, when $A\beta_{40}$ was mixed with A$\beta$(25–35), very small change in cell viability was observed. Our data indicated that our designed bipartite peptides might have therapeutic potential in amyloid-induced toxicity.

### Therapeutic effect of $R_8$-A$\beta$(25–35)-PEI in *APP/PS1* transgenic mice

To test whether $R_8$-A$\beta$(25–35) could prevent the deterioration of memory *in vivo*, PEI or PEI-coupled $R_8$-A$\beta$(25–35), denoted as $R_8$-A$\beta$(25–35)-PEI, was given intranasally for 4 months to *APP/PS1* mice when they were 4 months of age (experimental sets 1 and 2, Appendix Fig S3). The water maze assay was performed when the mice reached 8 months of age. As shown in Fig 3A, the wild-type mice treated with PEI or $R_8$-A$\beta$(25–35)-PEI showed no clear difference in the learning curve of finding the hidden platform. In contrast, the peptide-treated *APP/PS1* mice exhibited a significant

improvement in learning compared to the control transgenic mice treated with PEI. In addition, peptide-treated *APP/PS1* mice performed better at the probe test, as evidenced by their higher crossing number (Fig 3B) and longer time spent in the quadrant where the probe used to be compared to PEI-treated control transgenics (Fig 3C).

We next assessed the changes in the level of A$\beta$ peptide in the experimental set 1 animals by ELISA. As shown in Fig 3D and E, at the age of 8 months, the level of $A\beta_{40}$ and $A\beta_{42}$ decreased by 73% and 60%, respectively, in the hippocampus of peptide-treated *APP/PS1* mice compared with those of PEI-treated transgenic mice (Fig 3D). Similarly, the level of $A\beta_{40}$ and $A\beta_{42}$ decreased by 86% and 32%, respectively, in the cortex of the former group compared with the latter (Fig 3E). Our data indicate that the peptide treatment effectively reduced A$\beta$ accumulation and slowed down the clinical impairment of memory. Amyloid deposition is known to induce neuroinflammation, which contributes to disease pathogenesis in these mice. We therefore conducted cytokine assays. $R_8$-A$\beta$(25–35)-PEI effectively decreased the level of pro-inflammatory cytokines interleukin (IL)-6 and IL-1$\beta$ in the cortex (Fig 3F) in parallel with the changes in the level of A$\beta$ peptides.

### Continuous therapeutic effect of $R_8$-A$\beta$(25–35)-PEI after a suspension for 4 weeks

During the water maze tests, the treatment was adjourned for about 4 weeks. To examine whether the therapeutic effect could be maintained after a suspension of treatment, administration of PEI or

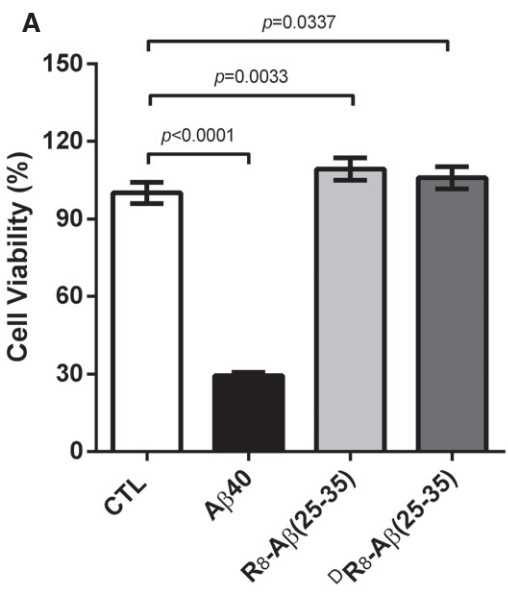

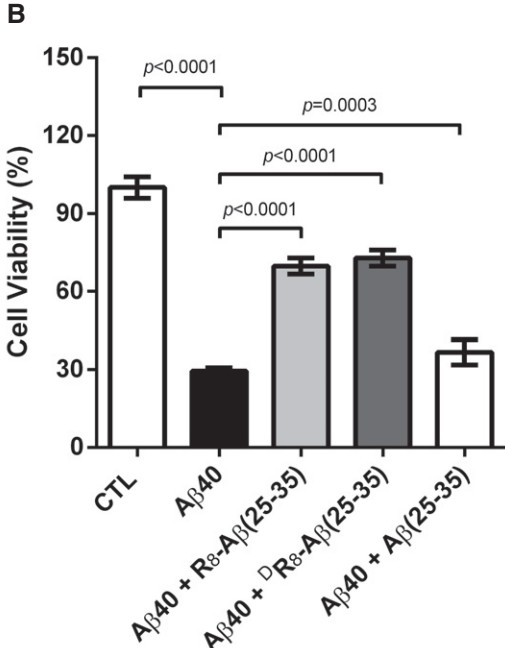

**Figure 2. Cell viability measurement by MTT assays.**

A   Neuro2a cells treated with DMSO (control), $A\beta_{40}$, $R_8$-A$\beta$(25–35), or $^DR_8$-A$\beta$(25–35).
B   Neuro2a cells treated with DMSO (control), $A\beta_{40}$, and $A\beta_{40}$ with equal molar $R_8$-A$\beta$(25–35), $^DR_8$-A$\beta$(25–35), or A$\beta$(25–35).

Data information: Peptide concentration is 30 μM for each peptide. Standard deviations of the mean are shown as bars for each sample ($N = 6$ for pure peptide and $N = 12$ for mixture). The statistics were done by Student's $t$-test.

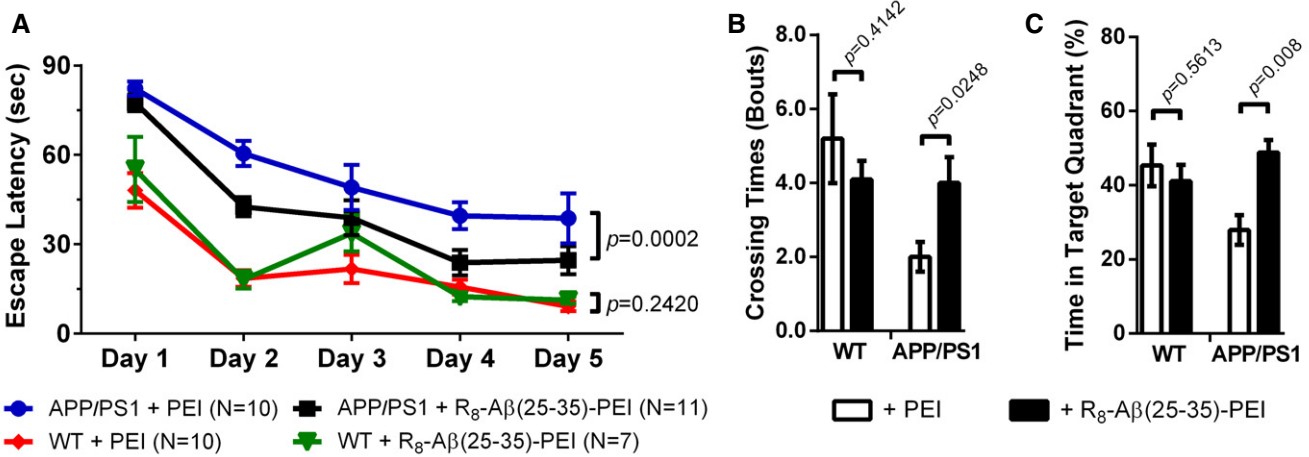

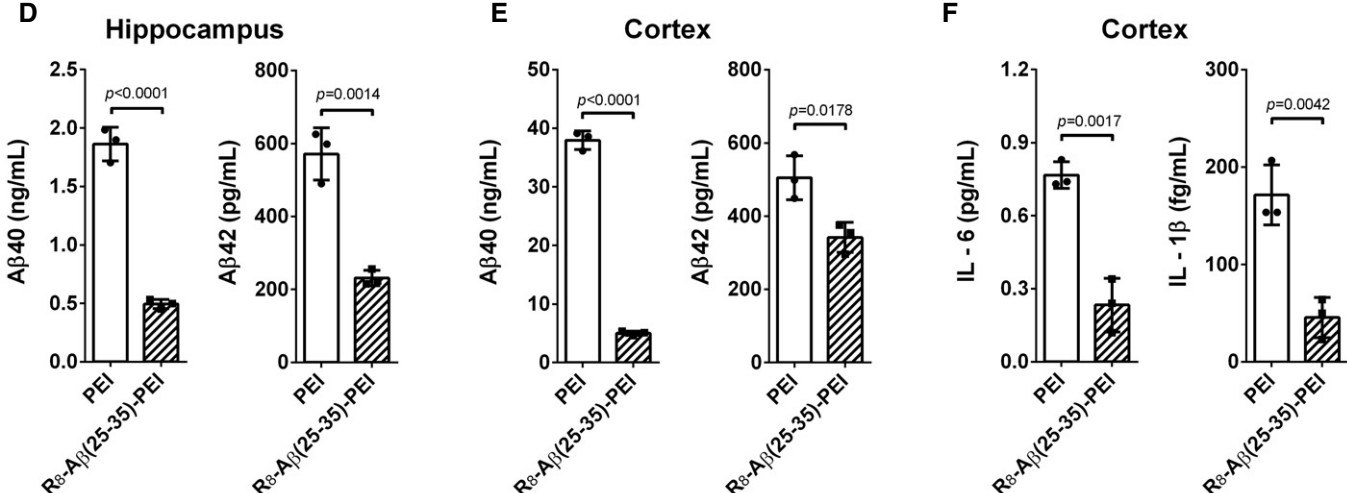

**Figure 3.  Effect of intranasally delivered R$_8$-Aβ(25–35)-PEI on *APP/PS1* transgenic mice after 4-month treatment.**

Wild-type (WT) and *APP/PS1* mice were treated with either PEI or R$_8$-Aβ(25–35)-PEI from the age of 4 months to 8 months.

A–C  Morris water maze. (A) The plot of the escape latency. (B) The times of the indicated mice crossing the target quadrant. (C) Percentage of time the indicated mice spent swimming in the target quadrant where the hidden platform used to be. The behavior data were expressed in mean ± SEM. The statistics of the escape trend were done with two-way ANOVA. Others were done by Student's *t*-test.

D, E  ELISA of total Aβ$_{40}$ and Aβ$_{42}$ in hippocampus (D) and cortex (E) ($N$ = 3 per group).

F      Level of IL-6 and IL-1β in the cortex ($N$ = 3 per group).

Data information: The data were expressed in mean ± SD, and the statistics were done by Student's *t*-test for panels (D–F).

peptide was resumed and continued for 4 more months (experimental set 2, Appendix Fig S3). The accumulation of amyloid plaques was quantified with microPET using the tracer Pittsburg compound B (PiB). As shown in Fig 4A and B, the peptide-treated *APP/PS1* mice had a much lower signal in the cortex, hippocampus, amygdala, and olfactory bulb compared with the PEI-treated mice, consistent with a beneficial therapeutic effect at this age. ELISA analyses revealed a significant decrease in SDS-insoluble Aβ$_{40}$ and Aβ$_{42}$ by 25–30% in the cortex or hippocampus of the peptide-treated *APP/PS1* mice compared with those in PEI-treated mice (Fig 4C and D), consistent with the microPET results (18–33% reduction). Correspondingly, SDS-soluble Aβ$_{40}$ and Aβ$_{42}$ levels significantly increased after peptide treatment (Fig 4E and F). One important biomarker in AD diagnosis

is the decrease in Aβ$_{42}$ level in the cerebral spinal fluid due to Aβ aggregation; a reversion of this process is expected to increase soluble Aβ concentration. Thus, these results demonstrated that our inhibitor peptide was able to inhibit Aβ from self-associating into amyloid fibrils/plaques.

**Inhibition effect of R$_8$-Aβ(25–35)-PEI on performed amyloid plaques in older mice**

To test whether R$_8$-Aβ(25–35)-PEI could prevent Aβ accumulation when amyloid plaques had already formed, we started peptide treatment with higher dosage (4 nmole/mouse/day) in another set of mice from the age of 8 months for 2 months until they were 10 months of

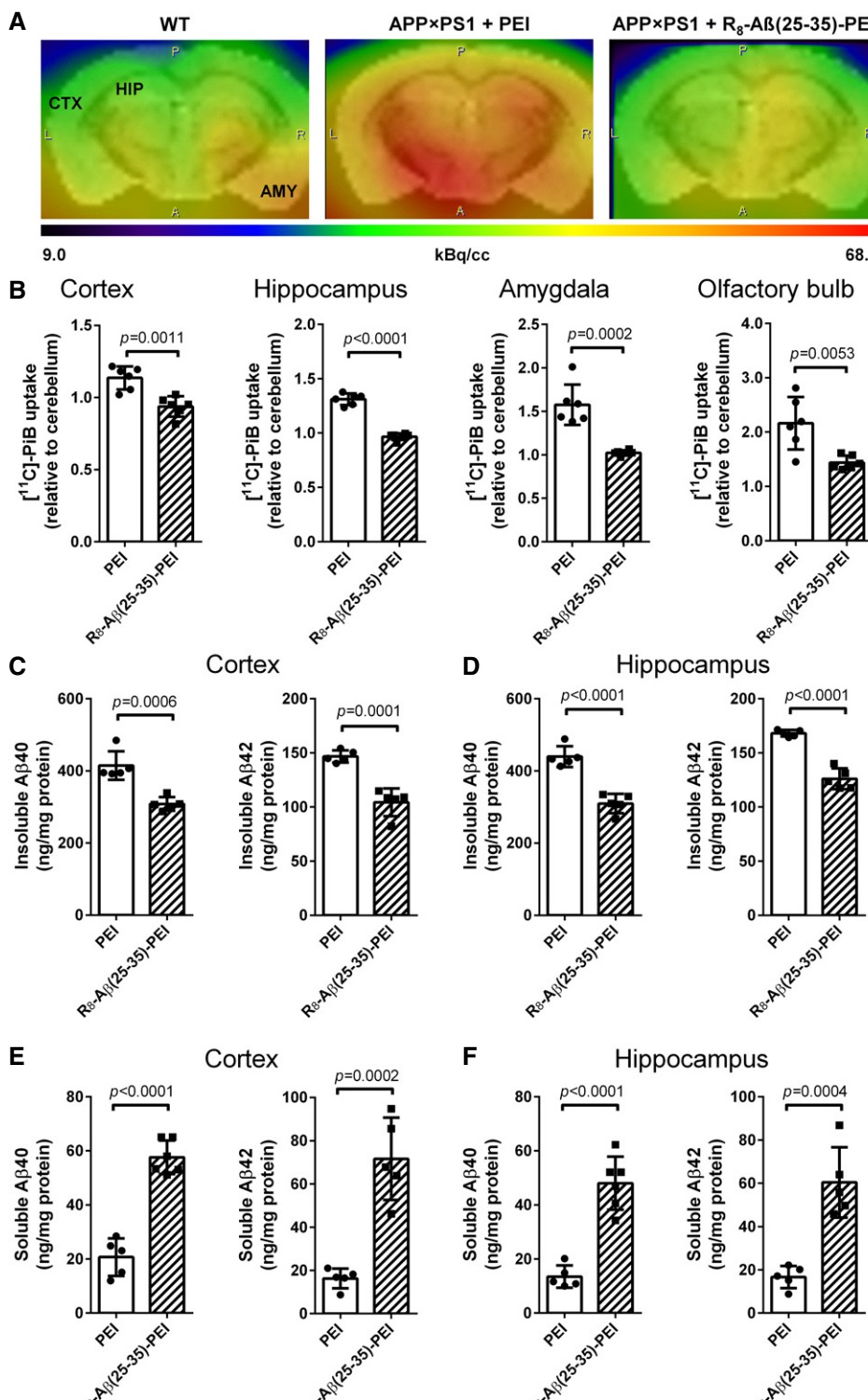

**Figure 4. Effect of intranasally delivered $R_8$-Aβ(25–35)-PEI on *APP/PS1* mice from 4 months to 13 months of age with a 1-month break within this period.**

A   Representative MicroPET image of the transgenic mouse brains co-registered with mouse T2-weighted MRI brain template. CTX, cortex; HIP, hippocampus; AMY, amygdala.

B   Quantitation of [$^{11}$C]PiB uptake in the cortex, hippocampus, amygdala, and olfactory bulb ($N$ = 6 per group).

C–F   ELISA of SDS-insoluble $Aβ_{40}$ and $Aβ_{42}$ in the cortex (C) and hippocampus (D) and SDS-soluble $Aβ_{40}$ and $Aβ_{42}$ in the cortex (E) and hippocampus (F) ($N$ = 5 per group).

Data information: Data were expressed in mean ± SD, and the statistics were conducted with the Student's *t*-test.

age (experimental set 3 in Appendix Fig S3). As shown in Appendix Fig S4, treatment with $R_8$-A$\beta$(25–35)-PEI did not decrease the number of amyloid plaques, but significantly decreased several parameters, including the size of individual plaques, the percentage of the cortex covered by plaques, and the total area of amyloid plaques per cortex, by 20%, 22%, and 24%, respectively. These data confirmed the therapeutic effect of this peptide even when administrated for a short time in mice with pre-existing amyloid plaques.

To investigate whether the therapeutic effect of $R_8$-A$\beta$(25–35)-PEI was possible with the Arg-rich segment alone, without needing A$\beta$(25–35), we also conducted experiments with $R_8$-YS-PEI peptide in the set 3 mice. No therapeutic benefit was observed (Appendix Fig S4). The results confirmed that $R_8$-A$\beta$(25–35)-PEI required the A$\beta$(25–35) segment for target recognition to reduce the amyloid accumulation.

### Entrance of $R_8$-A$\beta$(25–35)-PEI peptide into brains

To determine whether the intranasally given peptide inhibitor entered brains, we synthesized fluorescence-conjugated peptide, FITC-(Ahx)-C$R_8$-A$\beta$(25–35)-PEI. A higher dosage (5 µl of 1,800 µM peptide; 9 nmole/mouse/day) was given daily to one 10-week-old female C57BL/6JNarl mouse for three consecutive days in these tests in order to enhance the success rate of detection (experimental set 4 in Appendix Fig S3). Brains were collected and processed at 0.5, 2, 6, 12, and 24 h after the completion of the $3^{rd}$ peptide treatment. The amount of the intracerebral peptide was quantified by the FITC emission spectra between 500 and 600 nm of the brain filtrates excited at 446 nm (Fig 5A) against a calibration curve (Appendix Fig S5). The peptide was indeed detectable in the brains, which reached the peak (5.16 nmole) 6 h after the treatment, and then decreased at a rate about 0.086 nmole per hour (Fig 5B). In addition, the amounts of the peptide at the time points of 0.5 and 12 h were higher than that of 24 h. These data indicated that the peptide entered brain efficiently and was maintained at higher level for more than 12 h.

## Discussion

In this study, we demonstrated that the peptide $R_8$-A$\beta$(25–35) reduced the formation of amyloid fibrils by A$\beta_{40}$ *in vitro*, as well as amyloid plaques and disease manifestation *in vivo*. In a companion study, therapeutic peptides designed by the same modular principle also delayed disease in the R6/2 transgenic mice, a widely used mouse model for Huntington's disease (unpublished data). Thus, our data illustrated the possibility that this principle may be extended to design therapeutic peptides for other neurodegenerative diseases.

A variety of therapeutic peptides to decrease the formation of amyloid fibrils has been proposed (Funke & Willbold, 2012); our bipartite design works by attaching a polyR stretch to the peptide sequence derived from the disease-specific pathogenic peptide/ protein prone to aggregation. This approach possessed several unique features and advantages. First, the sequence directly taken from the pathogenic peptide/protein not only significantly reduced the labors of finding and optimizing a suitable peptide sequence, but also guaranteed high affinity with the target through its self-

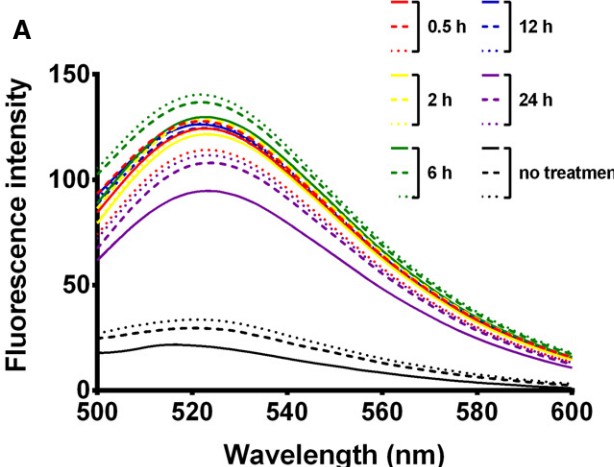

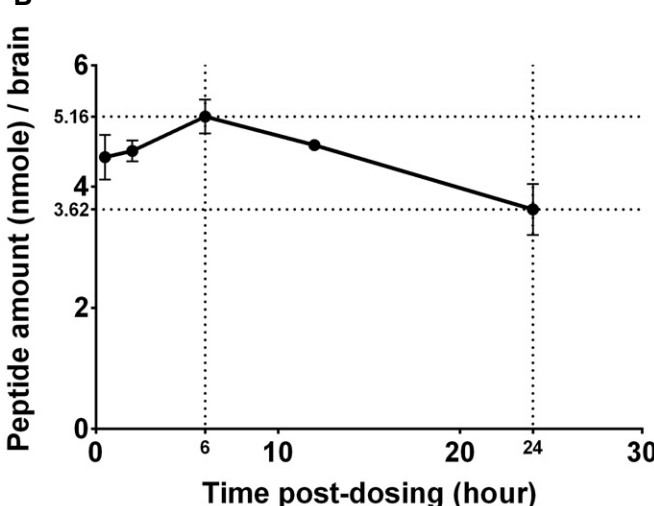

**Figure 5. Brain penetration of FITC-(Ahx)-C$R_8$-A$\beta$(25–35)-PEI after the third dosing via intranasal route.**

The mice were treated intranasally with FITC-(Ahx)-C$R_8$-A$\beta$(25–35)-PEI (9 nmole/ mouse/24 h) for three times. The mice were sacrificed, and their brains were perfused at 0.5, 2, 6, 12, and 24 h after the third treatment.

A  Fluorescence spectra of the filtrates of mouse brain homogenates after passing through a 100-kDa filter. The unbroken, dashed, and dotted lines represent three different mice.

B  The amount of FITC-(Ahx)-C$R_8$-A$\beta$(25–35)-PEI per brain at different times after the third peptide treatment. Data were expressed in mean $\pm$ SD ($n = 3$).

aggregating property. Second, the multi-charges in polyR rendered the designed therapeutic peptide (i) soluble in an aqueous environment and therefore simplifying the processes of synthesis and subsequent application, (ii) cell-penetrable (Mitchell *et al*, 2000), making it suitable for both extracellular and intracellular peptide/protein aggregation, and (iii) able to slow down oligomer/amyloid formation by charge repulsion after its binding to the pathogenic peptide/ protein. Third, combination of the polyR with the sequence from disease-specific pathogenic protein/peptide provided great feasibility and flexibility in applying this design across different misfolded aggregate-associated diseases.

Although many therapeutic peptides have been designed, only a few of them were tested *in vivo* (Permanne *et al*, 2002; van Groen *et al*, 2008; Frydman-Marom *et al*, 2009; Funke *et al*, 2010; Shukla *et al*, 2013; Lin *et al*, 2016). In this study, we have demonstrated the feasibility of intranasal administration of therapeutic peptidic prodrugs. When combined with technology in delivery, our study showed a proof of therapeutic principle for neurodegenerative diseases through intranasal delivery. The dose used in this study was only 2 nmoles (6 μg) per day, which was quite low compared with previous studies (Permanne *et al*, 2002; van Groen *et al*, 2008; Frydman-Marom *et al*, 2009; Funke *et al*, 2010). Using this dosage, we attempted to investigate the level of the therapeutic peptide in the brain during consecutive intranasal treatment (experimental set 5 in Appendix Figs S3 and S6). However, the peptide concentration was low and could not be reliably detected. As shown in Fig 5, after three consecutive treatments at higher amount (9 nmoles), there was 5.16 nmole of the peptide in the brain at 6 h after the final treatment and 3.62 nmole of the peptide in the brain 24 h after the final treatment. Although the current method was geared toward maximizing our ability to detect the intracerebral peptide rather than producing an accurate number in its efficiency in brain entrance, an estimated value was still achievable. Since the treatment continued for 3 days, the amount of intracerebral peptide before the $3^{rd}$ dose was expected not to be more than 3.62 nmole observed 24 h after the $3^{rd}$ treatment. Thus, at least 1.54 nmole (5.16 minus 3.62) or 17% of the daily dose of 9 nmole peptide entered brain. These results indicate that this peptide had a reasonably high therapeutic efficacy. Future studies will be conducted for optimal dosage.

The peptide treatment did not significantly decrease the numbers of the ThS-positive amyloid plaques, but reduced the size of the individual plaques and the total area of these plaques. One possibility is that most of the Aβ reduction is diffusely deposited Aβ. Alternatively, when we started treatment, the cores of plaques might have already formed at 4 months, but our peptide slowed down the speed of the accumulation of the transgenic Aβ of these plaques. Moreover, when we quantified SDS-soluble Aβ and SDS-insoluble Aβ separately (Fig 4C–F), we found that SDS-insoluble Aβ reduced after peptide treatment whereas SDS-soluble Aβ increased. Aβ accumulation is due to the imbalance of Aβ production and Aβ degradation. Our peptide treatment likely functions to inhibit Aβ from self-association, but may not directly impact on the Aβ degradation rate. The clearance of excessive Aβ depends on several Aβ-degrading enzymes, such as neprilysin (the most important one) and insulin-degrading enzyme, which were found to be downregulated in old mice (Caccamo *et al*, 2005). However, by preventing Aβ from aggregation, our peptide could render it more accessible to these Aβ-degrading enzymes and/or other degradation machinery in the brain. Recently, it has been reported that polyhydroxycurcuminoids upregulate neprilysin in the brain (Chen *et al*, 2016). Combining the peptide inhibitor and the neprilysin activator might additively enhance Aβ clearance.

Comparing peptide therapy and antibody therapy, the cost of peptide synthesis is much lower than the cost of producing monoclonal antibody. Moreover, as the peptide worked *in vivo* without incorporating non-natural or D-form amino acid, there was no worry for the toxicity caused by non-natural amino acids. Consistent with this, the preliminary tests for liver and kidney function indicated no clear toxicity in the mice receiving the peptide for 8 months (Appendix Fig S7).

Lastly, to determine whether the peptide treatment induced an antibody response against Aβ peptide, the serum of the mice treated for 15 days was tested and showed no evidence of immunoreactivity against the peptide (experimental set 6 in Appendix Figs S3 and S8). In summary, intranasal administration of our bipartite peptide designed on the principle of modular combination may serve as an effective and user-friendly disease-modifying therapy for Alzheimer's disease and a template for developing effective therapy against other protein aggregation-associated diseases.

# Materials and Methods

### Peptide synthesis

The peptides were prepared by the batch fluorenylmethoxycarbonyl (fmoc)-polyamide method (Lin *et al*, 2014). The sequence of Aβ40: DAEFRHDSGYEVHHQKLVFFAEDVGSNKGAIIGLMVGGVV. The sequence of $R_8$-Aβ(25–35): RRRRRRRRGSNKGAIIGLM; $^D R_8$-Aβ(25–35) had the same sequence as $R_8$-Aβ(25–35), but L-form arginines were replaced by D-form arginines. The sequence of $R_8$-YS: RRRRRRRRYS. The C-terminal ends of $R_8$-Aβ(25–35) and $^D R_8$-Aβ(25–35) were amidated by using Rink Amide AM resin (Novabiochem, Billerica, MA, USA) as the solid support. The synthesized peptides were cleaved from resin according to literature (Lin *et al*, 2014), purified by high-performance liquid chromatography (HPLC) using a Vydac C18 column, identified by matrix-assisted laser desorption ionization (MALDI) mass spectroscopy, and then lyophilized and stored at −20°C. To synthesize PEI-conjugated peptides $R_8$-Aβ(25–35)-PEI and $R_8$-YS-PEI, PEI was conjugated to the C-terminal carboxyl group of the peptides. Therefore, Fmoc-Met-Wang resin and Fmoc-Ser-Wang resin (Anaspec, USA) were used as the solid support during peptide synthesis. To prevent exopeptidase digestion *in vivo*, the N-terminal end of all the peptides in this study was acetylated.

### Circular dichroism spectroscopy

The 1.4 mM stock solution of each peptide in 75% trifluoroethanol was diluted to a final concentration of 30 μM in 20 mM sodium phosphate buffer with 150 mM KCl (PBS, pH 7). Each was incubated at 25°C for designated time points and placed in a 1-mm cell to record the CD spectra between 200 and 250 nm on a J-715 CD spectrometer (JASCO, Japan). The band width was set to 2 nm, and the step resolution was 0.05 nm. Each sample was scanned twice and the average of these two measurements was smoothed by the Savitzky–Golay method to get the final CD spectrum.

### Transmission electron microscopy

The samples were deposited on carbon-coated 300-mesh copper grids, incubated for 3 min for absorption, and then washed by water. Negative staining was carried out by staining with 2% uranyl acetate for 1.5 min. After air drying, the samples were viewed using a Hitachi H-7000 electron microscope (Hitachi, Japan).

## Cell viability assay

Mouse N2a neuroblastoma cells (ATCC) were cultured in Dulbecco's modified Eagle's medium (DMEM) (HyClone, USA) supplemented with 10% fetal bovine serum (HyClone, USA). Cells were harvested with DMEM and suspended at a density of $3.5 \times 10^5$ cells/ml. 100 μl from each sample was plated in one well of a 96-well CellBIND microplate (Corning, USA) and then incubated at 37°C under 5% $CO_2$ for 24 h. Five microliters from each stock peptide (6 mM) in DMSO was diluted with 95 μl of PBS (pH 7.0) and then 900 μl of DMEM to a final concentration of 30 μM. For testing efficacy of peptide inhibitor, $A\beta_{40}$ was premixed with equal volume of PBS or peptide as indicated, and incubated for 24 h at room temperature with shaking (50 rpm) before being added to the cultures. Viability was determined using the MTT (3-[4,5-dimethylthiazol-2-yl]-2,5-diphenyltetrazolium bromide) assay (Shearman *et al*, 1995) 48 h later as described in the literature (Chang *et al*, 2009).

## Synthesis of PEI-conjugated peptide

Three milligrams acetylated $R_8$-$A\beta$(25–35) was dissolved in 2.5 ml DMSO and slowly mixed with 150 μl 1-ethyl-3-(3-dimethylamino-propyl)carbodiimide (EDC) (600 mM in 0.1 M MES/0.5 M NaCl, pH 6) and 150 μl *N*-hydroxysuccinimide (NHS) (1.2 M in 0.1 M MES/0.5 M NaCl, pH 6). The mixture reacted at room temperature for 30 min with gentle shaking (70 rpm); then, 180 μl polyethylenimine (PEI) was added and reacted under the same conditions overnight. The PEI-conjugated peptide was purified by HPLC, lyophilized, and stored at −20°C.

## Intranasal administration

The animal experiments were approved by the Institutional Animal Care and Use Committee of the Academia Sinica. The methods were carried out in accordance with the approved guidelines.

*APP/PS1* (B6C3-Tg(APPswe,PSEN1dE9)85Dbo/Mmjax) trans-genic mice were purchased from Jackson Laboratories (Bar Harbor, Maine, USA) and maintained as described (Borchelt *et al*, 1996; Jankowsky *et al*, 2001). PEI, $R_8$-$A\beta$(25–35)-PEI, and $R_8$-YS-PEI were dissolved in 100 mM $NaH_2PO_4$/138 mM KCl (pH 5). The animal experimental designs are shown in Appendix Fig S3. In experiment 1, 2.5 μl PEI or $R_8$-$A\beta$(25–35)-PEI (400 μM) was given daily to each nostril of 4-month-old *APP/PS1* mice (eight female mice per group; 2 nmole/mouse/day) for 6 days per week until they were 8 months of age.

In experiment 2, 2.5 μl PEI or $R_8$-$A\beta$(25–35)-PEI (400 μM) was given daily to each nostril of 4-month-old *APP/PS1* mice (10 or 11 male mice per group; 2 nmole/mouse/day) for 6 days per week for a total of 8 months, with a suspension of 4 weeks for the Morris water maze test at 8 months of age. The same treatment was applied to non-transgenic littermates (10 male mice for PEI treatment and seven male mice for $R_8$-$A\beta$(25–35)-PEI treatment).

In experiment 3, 2.5 μl $R_8$-$A\beta$(25–35)-PEI (800 μM) was given daily to each nostril of three female *APP/PS1* mice (4 nmole/mouse/day) for 6 days per week from 8 to 10 months of age. To test the effect of $R_8$ peptide, three female *APP/PS1* mice were given

$R_8$-YS-PEI (800 μM), and three male *APP/PS1* mice were given buffer with the same intranasal method from 3 to 10 months of age (4 nmole/mouse/day; 6 days per week).

In experiment 4, 2.5 μl FITC-(Ahx)-$CR_8$-$A\beta$(25–35)-PEI (1,800 μM) was given daily to each nostril of 15 female C57BL/6JNarl mice (10 weeks old) for 3 days (9 nmole/mouse/day). In experiment 5, 2.5 μl FITC-(Ahx)-$CR_8$-$A\beta$(25–35)-PEI (400 μM) was given daily to each nostril of 33 female C57BL/6JNarl mice (12 weeks old) at day 1, day 2, and day 4 (2 nmole/mouse/day).

In experiment 6, 2.5 μl FITC-(Ahx)-$CR_8$-$A\beta$(25–35)-PEI (800 μM) or buffer was given daily to each nostril of six male non-transgenic littermates (1-year-old) for 15 days (three mice per group; 4 nmole/mouse/day).

## ELISAs for $A\beta_{40}$ and $A\beta_{42}$

The total levels of $A\beta_{40}$ and $A\beta_{42}$ in the brain homogenate of 8-month-old mice were detected using ELISA kits (Invitrogen, MD, USA) according to the manufacturer's instructions. Briefly, the cortical or hippocampal tissue was homogenized in ice-cold cell extraction buffer provided in the kit with protease inhibitor cocktail (Sigma, St. Louis, MO, USA) and centrifuged at 15,000 *g* at 4°C for 10 min. In the protocol, 5 M GdnHCl was used in $A\beta$ extraction buffer.

Because senile plaques started to form in mice older than 8 months of age, $A\beta$ in these plaques might not be extracted by SDS or GdnHCl (Kawarabayashi *et al*, 2001). $A\beta$ was separated into the SDS-soluble and SDS-insoluble fractions. Formic acid (FA) was used instead to extract SDS-insoluble $A\beta_{40}$ and $A\beta_{42}$ in plaques from the brain homogenate of the 13-month-old mice (van Groen *et al*, 2013). The brain tissues were first homogenized in the $A\beta$ extraction buffer containing 20 mM Tris–HCl (pH 7.6), 137 mM NaCl, 1% Triton X-100, 2% SDS, and protease inhibitor cocktail and centrifuged at 20,000 *g* for 20 min at 4°C. The super-natant was the SDS-soluble fraction for ELISA measurement of soluble $A\beta$. The pellet was then dissolved in 70% FA, sonicated for 1 min, and then centrifuged at 20,000 *g* for 20 min at 4°C. The resultant supernatant was the SDS-insoluble fraction. This fraction was neutralized with 20 volumes of 1 M Tris base before ELISA measurement. Total protein concentrations of the SDS-soluble and SDS-insoluble fractions were quantified using the Bradford protein assay (Bio-Rad, Hercules, CA, USA). $A\beta$ amount of each fraction was normalized to the total protein concentration of that fraction for comparison.

## Cytometric bead array

The levels of IL-6 and IL-1$\beta$ in cortical and hippocampal lysates were detected using Cytometric Bead Array (BD Biosciences, San Jose, CA, USA) according to the manufacturer's instructions and analyzed on the FACS Calibur (BD Biosciences, USA). The levels were calculated using CBA software (Soldan *et al*, 2004).

## Thioflavin S staining

Paraformaldehyde-fixed brain sections were applied with 1% (w/v) thioflavin S (ThS) solution for 10 min at room temperature protected from light, and then washed with 80% ethanol and water

to remove excessive dye. ThS-positive signals were visualized under an epifluorescence microscope; the plaque number, plaque area, and plaque size were analyzed by ImageJ (NIH, Bethesda, MD, USA).

## Morris water maze

The maze was conducted with visual cues on the wall and a hidden platform 1 cm beneath the water. The acquisition trial phase consisted of five training days (days 1–5) and four trials per day with a 15-min inter-trial interval. Mice were put into the maze from a different point in each trial. The path length and escape latency were recorded ($n = 7$–11 per group) and analyzed by two-way repeated-measures ANOVA. To assess their spatial memory, the platform was removed after three trials on day 5 and animals were allowed to swim freely for 90 s. The swimming path was recorded and analyzed.

## In vivo small-animal positron emission tomography imaging

All PET scans were performed using the Triumph pre-clinical tri-modality (LabPET/X-SPECT/X-O CT) imaging system (TriFoil Imaging, USA), which provides 31 transaxial slices 1.175 mm (center-to-center) apart, a 100-mm transaxial FOV, and a 37-mm axial FOV for the LabPET sub-system. The digital APD detector technology delivers high spatial resolution better than 1 mm and high recovery coefficient. Before the scans, all of the mice were kept warm with a heating lamp. After induction with 2.0% isoflurane, the mice were placed with their heads in the center of the field of view and were fixed in prone position. A 20-min static data acquisition was performed in 3D list mode with an energy window of 350–650 keV at 20 min following a [11C]PiB ($36.7 \pm 2.6$ MBq; volume < 0.25 ml) injection via the tail vein. The emission data were normalized and corrected for the tracer decay time. All list mode data were sorted into 3D sinograms, which were then single-slice Fourier rebinned into 2D sinograms. Summation images from 20 to 40 min after [11C]PiB injection were reconstructed using a MLEM algorithm, resulting in the image volume consisted of $240 \times 240 \times 31$ voxels, each voxel with a size of $0.25 \times 0.25 \times 1,175$ mm$^3$.

## Radiosynthesis of [11C]PiB

The radiosynthesis of [11C]PiB was performed using [11C]methyl triflate according to the method described previously with minimal modification (Manook et al, 2012). Briefly, [11C]methyl bromide was produced by the multi-pass bromination of [11C] methane. Subsequently, [11C]methyl bromide was eluted from a trap and converted to [11C]methyl triflate by passing a preheated silver triflate column. [11C]methyl triflate was carried by a helium stream (20 ml/min) into 350 μl of anhydrous methylethylketone containing 1.5 mg of 2-(4′-aminophenyl)-6-hydroxybenzothiazole. After the trapping, the reaction mixture was heated to 75°C for 2 min, and then, 0.4 ml of HPLC mobile phase was added to the reaction mixture for HPLC purification. HPLC purification was performed on a Waters Bondapak column (10 μm, 7.8 mm ID × 300 mm) using a mobile phase of acetonitrile/0.01 M H$_3$PO$_4$ (40/60) at a flow rate of 5.0 ml/min. The radioactive fraction corresponding to [11C]PiB was collected in a bottle containing

30 ml of pure water and passed through a C18 Sep-Pak Plus cartridge, then washed with 10 ml of pure water and eluted with 1 ml of ethanol and 10 ml of sterile normal saline, and passed through a 0.22-μm sterile filter for quality analysis and animal experiments. Radiochemical purity was > 99% as determined by analytical HPLC. The specific activity was $152 \pm 52$ GBq/μmol at the end of synthesis.

## PET data analysis

All imaging data were processed and analyzed with PMOD 3.5 software package (Pmod Technologies, Zürich, Switzerland). The PET image dataset was converted to an absolute measure of radioactivity concentration (kBq/cc) using a phantom-derived calibration factor before being normalized to the injected dose (ID) of [11C]PiB and the body mass of the animal. This normalization enables the comparison of brain radioactivity concentration of animals of different weights. Static PET images were co-registered with mouse T2-weighted MRI brain atlas based on PMOD as anatomic reference. Image origins were set to Bregma (0, 0) according to the MRI atlas and used the atlas for VOI definition. [11C]PiB uptake was evaluated in the four volumes of interest, namely the cortex, the hippocampus, the amygdala, and the olfactory bulb. Standardized uptake values (SUV) were obtained for each VOI by dividing the mean [11C]PiB activity by the injection dose and the body weight (gram). Thereafter, regional [11C]PiB uptake in the target region was normalized by [11C]PiB uptake in the cerebellum, which was taken as the reference region (ratio to cerebellum) (Manook et al, 2012; Poisnel et al, 2012).

## Detection of intracerebral peptide inhibitor

FITC-(Ahx)-CR$_8$-Aβ(25–35)-PEI peptide having fluorescein isothiocyanate (FITC) incorporated at the N-terminus and PEI at the C-terminus of the peptide inhibitor and aminohexanoic acid (Ahx) inserted as a spacer between FITC and peptide was synthesized.

Fifteen 10-week-old female wild-type C57BL/6JNarl mice were treated intranasally with 5 μl of 1,800 μM FITC-(Ahx)-CR$_8$-Aβ(25–35)-PEI peptide, dissolved in 100 mM NaH$_2$PO$_4$/138 mM KCl (pH 5), for three consecutive days (9 nmole/mouse/day). Another three mice without receiving peptide treatment were used as negative controls. These mice were perfused with PBS at 0.5, 2, 6, 12, 24 h after the last treatment ($n = 3$ per group), and their brains were collected and homogenized in the buffer containing 10 mM HEPES, 1.5 mM MgCl$_2$, 10 mM KCl, 100 mM DTT, and 1 mM PMSF (pH 7.9) (5 ml/gram of brain) by sonication on ice (one pulse of 0.75 s for 2 min, UP200H, Hielscher, USA). The homogenate was centrifuged at 21,000 $g$ for 20 min. After centrifugation, the supernatant was passed through a microconcentrator (100K cutoff, Pall Gelman, USA). Each filtrate was adjusted with the homogenization buffer to make a final volume of 1.5 ml, and then, their fluorescence emission spectra (500–600 nm) were recorded on a fluorescence spectrophotometer (FP-750, Jasco, Japan) with excitation at 446 nm. The pathlength was 1 cm, and slit width was set to 5 nm for both excitation and emission. To generate the concentration calibration curve, another two mice without treatment were perfused, and their brains were collected and cut into

**The paper explained**

**Problem**
Alzheimer's disease is an incurable neurodegenerative disorder. Disease progression usually extends over a period of 4–20 years. There is currently no effective treatment to delay its progression.

**Results**
We propose a modular approach to design peptide inhibitor against amyloid propagation. The peptide inhibitor can be delivered conveniently via intranasal administration and is shown to reduce brain amyloid deposition and ameliorate cognitive decline in Alzheimer transgenic mice.

**Impact**
The peptide nasal spray can be used as an efficient and clinically amenable treatment to delay the onset of Alzheimer's disease. Furthermore, our peptide design concept can extended to other protein aggregate-associated diseases.

two hemispheres. The left hemisphere was mixed with 20 µl of 1,800 µM FITC-(Ahx)-CR$_8$-A$\beta$(25–35)-PEI peptide, but the right one, with 20 µl of buffer (100 mM NaH$_2$PO$_4$/138 mM KCl, pH 5). They were homogenized, centrifuged, and filtered as described above. The filtrates were diluted to a final volume of 1.5 ml. The filtrate containing the peptide was mixed with the peptide-free filtrate at different ratios. Then, the fluorescence emission spectra of these mixtures were recorded to build the standard calibration curve.

**Expanded View** for this article is available online.

## Acknowledgements
We thank Mr. Tai-Lang Lin and the Imaging Core Facility of the Institute of Cellular and Organismic Biology, Academia Sinica, Taiwan, for assistance in transmission electron microscopy. Mass identification and CD spectroscopy of the synthesized peptides were performed in the Biophysics Core Facility in the Institute of Biological Chemistry, Academia Sinica. We thank the technical assistance of Ms. Wen-Ting Lin and Ms. Yi-Jhen Feng of the Histopathology Core Facility in the Institute of Biological Chemistry, Academia Sinica. We thank Ms. Yu-Ling Hwang and the Peptide Synthesis Core Facility in the Institute of Biological Chemistry, Academia Sinica, for peptide synthesis. We thank Animal Core Facility in the Institute of Biological Chemistry, Academia Sinica, and National Laboratory Animal Center for mouse care and breeding. We thank the Taiwan Mouse Clinic (MOST 104-2325-B-001-011) funded by the National Research Program for Biopharmaceuticals (NRPB) at the Ministry of Science and Technology (MOST) of Taiwan for technical support in MicroPET experiment and serum analysis. C.W. Chang is supported by the grant VGH102A-035 (Veterans General Hospital).

## Author contributions
Y-SC, Z-tC, T-YL, CL, HC-HS, Y-HW, and C-WC conducted the study and analyzed the data; R-SL contributed analytical tools and analyzed data; RP-YC and P-hT designed research study, analyzed data, and wrote the manuscript.

## Conflict of interest
A PCT patent application (PCT/US15/41785) has been filed.

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
