## [Review Process File · EMBO Molecular Medicine]

An intranasally delivered peptide drug ameliorates cognitive decline in Alzheimer transgenic mice

Yu-Sung Cheng, Zih-ten Chen, Tai-Yan Liao, Chen Lin, Howard C.-H. Shen, Ya-Han Wang, Chi-Wei Chang, Ren-Shyan Liu, Rita P.-Y. Chen, Pang-hsien Tu

Corresponding authors: Rita Chen and Pang-hsien Tu, Academia Sinica

Review timeline:

Submission date:	31 May 2016
Editorial Decision:	14 July 2016
Revision received:	07 December 2016
Editorial Decision:	10 January 2017
Revision received:	24 January 2017
Editorial Decision:	27 February 2017
Revision received:	01 March 2017
Accepted:	03 March 2017

Transaction Report:

Editor: Roberto Buccione

1st Editorial Decision

14 July 2016

Thank you for the submission of your manuscript to EMBO Molecular Medicine.

We are sorry that it has taken longer than usual to get back to you on your manuscript. In this case we experienced unusual difficulties in securing willing and appropriate reviewers. Furthermore, we could not obtain an evaluation from a third reviewer. Since a further delay cannot be justified, after further internal discussion, I have decided to proceed based on the two available and consistent evaluations.

As you will see, both Reviewers point to significant, fundamental and somewhat overlapping issues that in aggregate, preclude publication of the manuscript in EMBO Molecular Medicine in its current form. I will not discuss each point in detail as they are clearly stated.

In brief, while the two reviewers recognize the potential of the paper they raise several doubts that include unclear and unconvincing Aβ assays, poor statistical treatment of the data, unsatisfactory data presentation and clarity and finally, lack of pharmacology on the R8- peptides.

We are prepared to consider a substantially revised submission, with the understanding that the Reviewers' concerns must be fully and completely addressed with additional experimental data where appropriate and that acceptance of the manuscript will entail a second round of review. The overall aim is to significantly upgrade relevance and conclusiveness.

Please note that it is EMBO Molecular Medicine policy to allow a single round of revision only and that, therefore, acceptance or rejection of the manuscript will depend on the completeness of your responses included in the next, final version of the manuscript.

EMBO Molecular Medicine now requires a complete author checklist (<http://embomolmed.embopress.org/authorguide#editorial3>) to be submitted with all revised manuscripts. Provision of the author checklist is mandatory at revision stage; The checklist is designed to enhance and standardize reporting of key information in research papers and to support reanalysis and repetition of experiments by the community. The list covers key information for figure panels and captions and focuses on statistics, the reporting of reagents, animal models and human subject-derived data, as well as guidance to optimise data accessibility. This checklist especially relevant in this case given the issues raised with respect to statistical treatment and animal numbers.

As you know, EMBO Molecular Medicine has a "scooping protection" policy, whereby similar findings that are published by others during review or revision are not a criterion for rejection. However, I do ask you to get in touch with us after three months if you have not completed your revision, to update us on the status. Please also contact us as soon as possible if similar work is published elsewhere.

Please note that we now mandate that all corresponding authors list an ORCID digital identifier. You may do so through our web platform upon submission and the procedure takes <90 seconds to complete. We also encourage co-authors to supply an ORCID identifier, which will be linked to their name for unambiguous name identification.

Should you find that the requested revisions are not feasible within the constraints outlined here and choose, therefore, to submit your paper elsewhere, we would welcome a message to this effect.

***** Reviewer's comments *****

Referee #1 (Remarks):

Interesting study using a peptide corresponding to A β sequence 25-35 with an N-terminal R8 extension and a C-terminal PEI tail for the inhibition of A β aggregation in vitro and in vivo (APP/PS1 mouse model). There is no control with R8 alone and no control with PEI alone. The MTT cytotoxicity assay with Neuro2a mouse neuroblastoma cells has no in vivo relevance. The terminology of "inclusion bodies (IBs)" - used throughout the paper - for amyloid/amyloid fibrils is not acceptable. The list of neurodegenerative disease with the so called inclusion bodies is incomplete and lacks prominent members. It is also not correct to state that all neurodegenerative diseases are in conjunction with "inclusion bodies" and that there is a gradual neuronal loss. The loss is found in certain regions of the brain (e.g. entorhinal cortex layer II).

It is unclear and not discussed that total brain A β 42 is not reduced in the treated animals whereas A β 40 is. Unclear, because amyloid density measured with amyloid-PET and thioflavin staining is reduced after treatment. Since A β 42 is the major constituent of amyloid plaques, the question arises whether the A β 42 assay were done correctly.

Additionally, intranasal treatment has been done with humans. The corresponding paper on intranasal insulin treatment of patients with Alzheimer's disease need to be cited and discussed. Also in context with the issue why PEI was used.

The term "aggregation-caused neurodegenerative diseases" is misleading and not in agreement of what is currently discussed as the cause of neurodegenerative disease with protein depositions.

Referee #2 (Comments on Novelty/Model System):

Relevant comments are included in the comments to the authors below.

Referee #2 (Remarks):

The manuscript entitled "An intranasally delivered peptide drug ameliorates the cognitive decline in Alzheimer transgenic mice" describes the generation and initial testing of a peptide inhibitor of Abeta aggregation. The peptide is based on sequences 25-35 of Abeta with the addition of 8 Arg residues to produce charge-based repulsive forces to block aggregation. The authors demonstrate that the peptide itself does not form the typical fibrillary aggregates of Abeta and that the peptide modulate the formation of fibrillary aggregates by Abeta 40. The authors report that administration of the peptide to APP^{swe}/PS1^{dE9} mice by intranasal delivery was effective in ameliorating cognitive behavioral deficits in male mice and attenuating the deposition of amyloid plaques. The work is potentially very interesting, but there are several aspects of the work that require some additional clarification.

Fig 1) The in vitro studies of Abeta 40 aggregation were performed at 1:1 molar ratios. For this peptide to really have any potential for translation to human, it would have to be more efficacious than shown here. What happens if the ratio of peptide to Abeta40 is 1:5 or 1:10? I am not sure one can interpret the data in Fig 1I as an indication that the kinetics of Abeta 40 aggregation were slowed. Please specify the exact time that TEM images were taken for Fig. 1J and 1K. Both the Results and legends indicate that incubation times for data in each panel should have been noted somewhere on the Figure. I do not see such indications. It is not clear to me that the R8-Ab-25-35 peptides have any inhibitory effect on Abeta 40 fibrillization. I can see plenty of fibrils in the TEM images. Finally, it is not clear why the authors chose to look at Abeta 40 aggregation in these studies, when all available data implicates Abeta 42 as the primary culprit.

Fig 2) I would much prefer to see the error bars for these data as Std Deviation rather than Std error of the mean if that is what has been used. I am not sure I understand how these data were analysed. Each bar is stated to be a generated from a triplicate experiment and the study was repeated 3 times. Are these bars the average of 9 data points or 3? Should really be 3 with each triplicate averaged to produce one data point per experiment.

Fig 3) It was not entirely clear to me how many animals were used to produce the Abeta measurements by ELISA and the pathology quantification in this Figure. I would much prefer to see the data in D, E, G, H, I, J, K displayed as scatter plots. Please provide a better description of how the statistical analysis was done. What does N=3 per group for D and E mean? Does this mean treatment group, and only 3 animals analyzed? Why not analyze all 8 animals available? For G and H, were only 2 animals analyzed? Or do you mean 2 sections were analyzed per animal? I don't understand how the statistics were done in G and H if only 2 animals. Are the graphs in G and H plaques per brain?

Fig 4) I would again like to see the data as scatter plots. I frankly have never seen such a low level of variation for these types of data.

A major element that is lacking in this paper is any pharmacology on the R8- peptides. What level of these peptides is achieved in the brain by the dosing regimen used? What is the half-life of these peptides in brain? Is it detectable in CSF after dosing? What level of peptide in brain is required to produce the therapeutic effect? Is there any antibody response to the peptide with repeated dosing?

1st Revision - authors' response

07 December 2016

RESPONSE TO COMMENTS OF REVIEWERS:

We would like to thank both reviewers for their insightful and constructive critics that address important issues/weaknesses in our original manuscript. In response, we have added additional studies and made corresponding and necessary changes, which include significant revisions of the four figures in the original manuscript, inclusion of , new Figure 5 in the revised manuscript, seven new supplemental figures and one method in the supplemental information, and 10 more references. These changes we believe have substantially strengthened our revised manuscript. For your convenience, we highlighted the changes in red in the revised manuscript. Below are the details of our responses to these comments.

Reviewer 1:***Comment 1: There is no control with R8 alone and no control with PEI alone***

Our response: In our original manuscript, the animal study was in fact conducted with PEI as control. It is described in the beginning of the section “**Therapeutic effect of R₈-A β (25-35)-PEI in APP/PS1 transgenic mice**” in the Results – “To test if R₈-A β (25-35) could prevent the deterioration of memory *in vivo*, PEI or PEI-coupled R₈-A β (25-35), denoted as R₈-A β (25-35)-PEI, was given intranasally starting from 4 months of age.” (please refer to page 11 of revised manuscript), in the method describing intranasal administration (page 22), and in Figures 3 and 4.

Indeed, we did not test the R₈ sequence, and agree with the reviewer that it is appropriate to incorporate it as a control. So we have conducted a new set of experiments treating the APP/PS1 mice with R₈-YS-PEI in response to the reviewer’s comment. The peptide contained additional amino acids Tyrosine and Serine to facilitate peptide purification because pure R₈ was too hydrophilic to be purified by C18 column (see the method on page 19 for details). The results are shown in the supplemental information Fig. S4. In short, R₈-YS-PEI did not show clear therapeutic benefit as did the R₈-A β (25-35)-PEI, suggesting the Ab sequence 25-35 is important for target recognition. See page 14. To avoid confusion, all sets of animal experiments used in this study are summarized in supplemental information Fig. S3.

Comment 2: The MTT cytotoxicity assay with Neuro2a mouse neuroblastoma cells has no in vivo relevance.

Our response: The MTT cytotoxicity assay using Neuro2a cells was a widely used convenient cellular platform to investigate a wide variety of biological properties and/or toxicity of proteins and drugs. It was chosen in this study to provide initial information on (1) any potential cytotoxicity of R₈-Ab(25-35) peptide and (2) its ability to decrease/efface toxicity of A β ₄₀. The acquired information was crucial for us to test our design of therapeutic peptide and its potential *in vivo* therapeutic value before moving toward the time- and resource-demanding animal studies. It was also used to check whether the D-form counterpart behaved similarly to the L-form peptide.

Comment 3: The terminology of "inclusion bodies (IBs)" - used throughout the paper - for amyloid/amyloid fibrils is not acceptable. The list of neurodegenerative disease with the so called inclusion bodies is incomplete and lacks prominent members. It is also not correct to state that all neurodegenerative diseases are in conjunction with "inclusion bodies" and that there is a gradual neuronal loss.

Our response: Here we would like to make some clarification first. We did not use the term inclusion bodies as a synonym for amyloid/amyloid fibrils; instead we intended to use it for the misfolded proteinaceous aggregates that are present in a variety of neurodegenerative diseases as mentioned in the manuscript. Although amyloid feature is prominent among the aggregates of these diseases, there are clear evidence that not all aggregates are amyloid, particularly the TDP-43-positive aggregates in a form of frontotemporal dementia and Amyotrophic lateral sclerosis (Robinson et al. Acta Neuropathol. 2013 January ; 125(1): 121–131. doi:10.1007/s00401-012-1055-8; Bigio et al. Acta Neuropathol. 2013 March ; 125(3): 463–465. doi:10.1007/s00401-013-1089-6.). Therefore, a more generic term like inclusion bodies or proteinaceous aggregates are preferred when referring to these abnormal pathologic structures across different diseases. However in response to the reviewer’s concern, we have changed the term “inclusion body” to “misfolded (proteinaceous) aggregates” (Abstract, p.3; Introduction, p.5-6; Discussion, p.17), and reserved amyloid/amyloid fibrils for the aggregates formed by A β in human Alzheimer’s disease and the mouse model.

We agree with the reviewer that the list of neurodegenerative diseases characterized by signature misfolded aggregates in our manuscript is incomplete for we only intended to mention some major entities as examples. So we have omitted others like prion diseases, Tau proteinopathies and a long list of others relatively rare diseases etc. However, in contrast with the impression stated by the reviewer, every neurodegenerative disease mentioned in our manuscript is in fact prominent and prototypic among its own disease category, such as Alzheimer’s disease, the most common neurodegenerative disease leading to dementia; Parkinson’s disease, the most common disease leading to motor deficits; Huntington’s disease, the prototype of a large category of trinucleotide repeat diseases; Amyotrophic lateral sclerosis, the most common disease inflicting the spinal cord, etc. All of these diseases are currently subjects of intensive research efforts.

Comment 4: It is unclear and not discussed that total brain Ab42 is not reduced in the treated animals whereas Ab40 is. Unclear, because amyloid density measured with amyloid-PET and thioflavin staining is reduced after treatment. Since Ab42 is the major constituent of amyloid plaques, the question arises whether the Ab42 assay were done correctly.

Our reply: We thank reviewer for this insightful and legitimate question, and concede that the way our data presented is both non-intuitive and confusing. In fact, the experiments were conducted by measuring the soluble and insoluble fractions of Ab40 and Ab42 with commercial ELISA kit; the total level was then calculated by combining the insoluble and the soluble pools of each A β species. In response to the comment, we reviewed our data, and discovered mistakes in our method to combine the soluble and insoluble Ab into total Ab with failures to include the normalizing factor like homogenization volumes. To avoid confusion, we replaced the data of total A β with those of the soluble A β in our revised Figs.4E and 4F, and revised our manuscript with the findings and relevant discussions (page 13):

“Correspondingly, soluble A β_{40} and A β_{42} levels significantly increased after peptide treatment (Fig. 4E & 4F). One important biomarker in AD diagnosis is the decrease of Ab levels in the cerebral spinal fluid due to Ab aggregation; a reversion of this process is expected to increase soluble Ab concentration. Thus, these results demonstrated that our inhibitor peptide was able to inhibit Ab from self-associating into amyloid fibrils/plaques. The clearance of excessive A β depends on several major Ab-degrading enzymes, such as neprilysin (the most important one) and insulin-degrading enzyme, which were found to be downregulated in old mice (Caccamo et al, 2005). By preventing A β from aggregation, our peptide could render it more accessible by these Ab-degrading enzymes and/or other degradation machinery in the brain. Recently it has been reported polyhydroxycurcuminoids upregulated neprilysin in the brain (Chen et al, 2016). Combining the peptide inhibitor and the neprilysin activator might additively enhance Ab clearance.”

Comment 5: Additionally, intranasal treatment has been done with humans. The corresponding paper on intranasal insulin treatment of patients with Alzheimer's disease need to be cited and discussed. Also in context with the issue why PEI was used.

Our reply: We thank the reviewer for the suggestion. We have cited the reference about intranasal insulin (Claxton A, Baker LD, Hanson A, Trittschuh EH, Cholerton B, Morgan A, Callaghan M, Arbuckle M, Behl C, Craft S (2015) Long-acting intranasal insulin detemir improves cognition for adults with mild cognitive impairment or early-stage Alzheimer's disease dementia. *J Alzheimers Dis* 44: 897-906; Craft S, Baker LD, Montine TJ, Minoshima S, Watson GS, Claxton A, Arbuckle M, Callaghan M, Tsai E, Plymate SR, Green PS, Leverenz J, Cross D, Gerton B (2012) Intranasal insulin therapy for Alzheimer disease and amnesic mild cognitive impairment: a pilot clinical trial. *Arch Neurol* 69: 29-38) and discussed on pages 7-8.

Comment 6: The term "aggregation-caused neurodegenerative diseases" is misleading and not in agreement of what is currently discussed as the cause of neurodegenerative disease with protein depositions.

Our reply: We agree with the reviewer that the role of aggregates as the pathogenic cause of the neurodegenerative diseases is still not yet settled. However, recently publications indicated that the misfolded aggregates possess prion-like characteristics and are detrimental have lent support to their importance (Guo and Lee, 2014. *Nat. Med.* 20,130–138; Prusiner et al., *PNAS*. 2015. 112 (38): E5308–E5317). However, in response to the reviewer's concern for potential confusion and controversy, we have changed the wording from “aggregation-caused” to scientifically neutral “aggregation-associated.” (Abstract, p.3; Discussion, p.17 & 19).

Reviewer 2

Comment 1: Fig 1) The in vitro studies of Abeta 40 aggregation were performed at 1:1 molar ratios. For this peptide to really have any potential for translation to human, it would have to be more efficacious than shown here. What happens if the ratio of peptide to Abeta40 is 1:5 or 1:10? I am not sure one can interpret the data in Fig 1I as an indication that the kinetics of Abeta 40 aggregation were slowed. Please specify the exact time that TEM images were taken for Fig. 1J and 1K. Both the Results and legends indicate that incubation times for data in each panel should have been noted somewhere on the Figure. I do not see such indications. It is not clear to me that

the R8-Ab-25-35 peptides have any inhibitory effect on Abeta 40 fibrillization. I can see plenty of fibrils in the TEM images. Finally, it is not clear why the authors chose to look at Abeta 40 aggregation in these studies, when all available data implicates Abeta 42 as the primary culprit.

Our reply: We appreciate these insightful questions raised by this reviewer, and agree that the original figure is somewhat confusing and incomplete. In response, we have conducted different sets of experiments with varying molar ratios and replaced the original TEM pictures with new ones to better illustrate the points.

(1) As requested by the reviewer, we tested the efficacy of R₈-Ab(25-35) at lower molar ratios. The negative ellipticity at 218 nm indicates the cross- β structure formation which is the structural feature of amyloid fibrils. Even at the concentration of one tenth of that of Ab40, we observed the inhibitory effect of R₈-Ab(25-35). A new set of Transmission Electron Microscopy (TEM) studies with different molar ratios of A β 40 over R₈-Ab(25-35) were also conducted with similar results. These findings of CD spectra and TEM are shown in the revised Fig. 1. In addition, we added a schematic illustration of the hypothetical inhibition model by R₈-Ab(25-35) in the revised Fig. 1A for readers to grasp the concept of our peptide design. See new Fig. 1, Fig. S1, Fig. S2, pages 8-10.

(2) In the kinetics of CD spectra, Ab40 started to form fibrils after ~100 hours; however, in the presence of either Ab40+R₈-Ab(25-35) or Ab40+^DR₈-Ab(25-35) at 1:1 ratio, it failed to form fibrils. Likewise, we observed abundant fibrils in Ab40 sample, but only scanty or no fibrils in A β 40/R₈-Ab(25-35) or A β 40/^DR₈-Ab(25-35). R₈-Ab(25-35) and ^DR₈-Ab(25-35) only formed amorphous aggregates after long time incubation. We agree with the reviewer that Figs. 1J, 1K were hard to interpret. As stated above, we have replaced them with new images in the revised Figure 1, which should clearly demonstrate the reduction in the Ab fibrils by the therapeutic peptide in a dose-dependent manner. Instead, the control data of R₈-Ab(25-35) and ^DR₈-Ab(25-35) were moved to supplemental Figs. S1 and S2 in the newly added supplemental information. Fig. S1 showed the CD spectra and TEM images of R₈-Ab(25-35) and ^DR₈-Ab(25-35). Fig. S2 compared the inhibition effect of R₈-Ab(25-35) and ^DR₈-Ab(25-35) on Ab40 fibrillization.

(3) Concerning using Ab42 in our studies, first of all, *in vitro* Ab40 has been widely used in many previous literatures to study Ab inhibitors (Lowe *et al. Biochemistry*, 2001, 40, 7882; Findeis *et al. Biochemistry*, 1999, 38, 679; Gordon *et al. Biochemistry* 2001, 40, 8237 and *J. Peptide Res.* 2002, 60, 37). Secondly, we did attempt to repeat the same studies with A β 42 peptide; however, our efforts were not successful due to repeated failures either in synthesis at our own facility or in fibrillization from the commercial vendors. Thirdly, we reasoned that *in vivo* efficacy of the peptide take priority over *in vitro* studies, we decided to proceed to the animal studies to investigate the *in vivo* benefit of our peptide with concomitant changes in A β 42 after enough *in vitro* data were gathered either biochemically or in cell culture with A β 40.

Comment 2: Fig 2) I would much prefer to see the error bars for these data as Std Deviation rather than Std error of the mean if that is what has been used. I am not sure I understand how these data were analysed. Each bar is stated to be a generated from a triplicate experiment and the study was repeated 3 times. Are these bars the average of 9 data points or 3? Should really be 3 with each triplicate averaged to produce one data point per experiment.

Our reply: (1) We revised all the figures and used standard deviation to express the error bars except the behavior data in which more mice were used. See revised Fig. 2-4 and new supplemental Fig. S4, Fig. S6 in supplemental information.

(2) For figure 2, we are sorry to mislead the Reviewer. We have done the cytotoxicity experiment three times independently to make sure the same conclusion can be acquired. We did not combine all the data but only show the results of the last time. We analyzed 6 wells for pure peptides and 12 wells for mixtures. The N value (how many wells of cells were treated and measured) is indicated in the figure legend for clarity. See page 38.

Comemnt 3: Fig 3) It was not entirely clear to me how many animals were used to produce the Abeta measurements by ELISA and the pathology quantification in this Figure. I would much prefer to see the data in D, E, G, H, I, J, K displayed as scatter plots. Please provide a better description of how the statistical analysis was done. What does N=3 per group for D and E mean? Does this mean treatment group, and only 3 animals analyzed? Why not analyze all 8 animals available? For G and H, were only 2 animals analyzed? Or do you mean 2 sections were analyzed per animal? I don't understand how the statistics were done in G and H if only 2 animals. Are the graphs in G and H plaques per brain?

Our reply: To make it clear, we summarize all the animal experiments in supplemental information Fig. S3. The mouse number (N) for each experiment is indicated in the figure legend now. For the 8-month-old female mice, we planned different experiments for these brains. We used 3 mice per group for Ab ELISA, 2 mice per group for ThS staining, and 3 mice per group for IL-6/IL-1b analysis. The results from different experiment are consistent. Since the Reviewer questioned that mice used in the ThS-staining results of the 8-month-old mice are too few, we have removed ThS staining results from Figure 3. See pages 11-12 and revised Figure 3. Instead, we have done one new set of animal experiment and used 3 mice for ThS staining (12 sections per mouse). In this animal experiment, we treated the APP/PS1 mice with R₈-Aβ(25-35)-PEI, PBS buffer, or R₈-YS-PEI. The R₈-YS-PEI peptide was composed of eight Arg, with one Tyr and one Ser (which were added to facilitate peptide purification) in order to exclude the possibility of efficacy of R₈-Aβ(25-35)-PEI came from the R₈ moiety. Please see the method section on pages 19 and 22-23. The results are shown in the supplemental information Fig. S4 and a new section was added on page 14:

“Inhibition effect of R₈-Aβ(25-35)-PEI on performed amyloid plaques in older mice

To test if R₈-Aβ(25-35)-PEI could prevent Ab accumulation when amyloid plaques had already formed, we started peptide treatment with higher concentration (800 μM) in another set of mice from the age of 8 months for 2 months until they were 10 months of age (experimental set 3 in supplemental Fig. S3). As shown in supplemental Fig. S4, treatment with R₈-Aβ(25-35)-PEI did not decrease the number of amyloid plaques, but significantly decreased several parameters, including the size of individual plaques, the percentage of the cortex covered by plaques and the total area of amyloid plaques per cortex, by 20%, 22% and 24%, respectively. These data confirmed the therapeutic effect of this peptide even when administrated for a short time in mice with pre-existing amyloid plaques.”

Comment 4: Fig 4) I would again like to see the data as scatter plots. I frankly have never seen such a low level of variation for these types of data.

Our reply: We agree with the comment, and, in response, have re-made the figures as scatter plots (see the revised Fig. 3 and Fig. 4). Indeed, animal data typically exhibit wider variation than what were presented in this manuscript. The microPET studies were independently conducted by the Imaging Core of the Taiwan Mouse Clinic, and experiments were done in a blind fashion because we would like to exclude any potential bias in favor of our expectation. The code was only broken after experiments for data analyses. Thus, it is logical for us to assume that the low level of variation may reflect the high level of intentional effort that had been taken while the Core performed those experiments. This Imaging Core has established experiences in microPET imaging (visit <http://tmc.sinica.edu.tw/micropet.html>).

Comment 5: A major element that is lacking in this paper is any pharmacology on the R8-peptides. What level of these peptides is achieved in the brain by the dosing regimen used? What is the half-life of these peptides in brain? Is it detectable in CSF after dosing? What level of peptide in brain is required to produce the therapeutic effect? Is there any antibody response to the peptide with repeated dosing?

Our reply: We agree with these constructive comments, and have conducted experiments to address these issues. .

1). Detecting intracerebral peptide. The experiments and results have been included in the revised manuscript (results p.15-16, method p.23, 28-29) and new Fig. 5. Below is the result from the revised manuscript.

“Entrance of R₈-Aβ(25-35)-PEI peptide into brains

To determine if the intranasally-given peptide inhibitor entered brains, we synthesized fluorescence-conjugated peptide, FITC-(Ahx)-CR₈-Aβ(25-35)-PEI. A higher dosage (5 mL of 1800 mM peptide; 9 nmole) was given daily to one 10-week-old female C57BL/6JNarl mouse for 3 consecutive days in these tests in order to enhance the success rate of detection. Brains were collected and processed at 0.5, 2, 6, 12, and 24 hours after the completion of the 3rd peptide treatment. The amount of the intracerebral peptide was quantified by the FITC emission spectra between 500 and 600 nm of the brain filtrates excited at 446 nm (Fig. 5A) against a calibration curve (supplemental Fig. S5). The peptide was indeed detectable in the brains, which reached the peak (5.16 mmole) 6 hours after the

treatment, and then decreased at a rate about 0.086 nmole per hour. There was still 3.62 nmole of the peptide in the brain 24 hours after treatment. Since the peptide was given at an interval of 24 hours, the amount of intracerebral peptide should be 3.62 nmole or more throughout the therapeutic period. The current method was geared toward maximizing our ability to detect the intracerebral peptide, rather than to produce an accurate number in its efficiency in brain entrance. However, an estimated value was still achievable. Since the treatment continued for 3 days, the amount of intracerebral peptide before the 3rd dose was expected not to be more than 3.62 nmole observed 24 hours after the 3rd treatment. Thus, at least 1.54 nmole (5.16 minus 3.62) or 17% of the daily dose of 9 nmole peptide entered brain. In addition, the amounts of the peptide at the 0.5 and 12 hours time points were higher than that of 24 hours. These data indicated that the peptide entered brain efficiently and was maintained at higher level for more than 12 hours.”

2) As for the antibody response, we have collected the mouse serum after peptide treatment. No specific antibody response can be found against A β peptide. Please refer to the revised manuscript (lines 5-8, p. 16) and the supplemental Fig. S7 with the method of dot blot assay (supplemental information p.9).

2nd Editorial Decision

10 January 2017

Thank you for the submission of your revised manuscript to EMBO Molecular Medicine.

We have now received the enclosed reports from the reviewers that were asked to re-assess it. As you will see the reviewers, while generally supportive, do have a number of remaining concerns for you to take action upon. Specifically, while reviewer 1 is asking you to correct a few imprecise statements, reviewer 2 remains puzzled by the definition of soluble Abeta in your manuscript and would also like to understand what the steady state level of the peptide in the brains might be. Finally, s/he suggests streamlining the results section.

Providing you deal with the above issues accurately and fully, I am prepared to make an editorial decision on your next, final version

In addition to appropriately dealing with the above issues, please introduce the following final editorial amendments:

- 1) Please correct the reference style. If you cannot find the EMBO Molecular Medicine template for your reference manager software, you can use the EMBO Journal one.
- 2) Please fully de-anonymise your manuscript text, including COI and Author Contribution sections
- 3) Please update manuscript callouts from Supplementary Figure 1, etc." to "Appendix Figure 1, etc.". The Appendix should be in a PDF format and should include a first page TOC and a header for the supplemental methods section.
- 4) As per our Author Guidelines, the description of all reported data that includes statistical testing must state the name of the statistical test used to generate error bars and P values, the number (n) of independent experiments underlying each data point (not replicate measures of one sample), and the actual P value for each test (not merely 'significant' or 'P < 0.05').
- 5) The manuscript must include a statement in the Materials and Methods identifying the institutional and/or licensing committee approving the experiments, including any relevant details (like how many animals were used, of which gender, at what age, which strains, if genetically modified, on which background, housing details, etc). We encourage authors to follow the ARRIVE guidelines for reporting studies involving animals. Please see the EQUATOR website for details: <http://www.equator-network.org/reporting-guidelines/improving-bioscience-research-reporting-the-arrive-guidelines-for-reporting-animal-research/>. Please make sure that ALL the above details are reported in the main text.
- 6) We encourage the publication of source data, with the aim of making primary data more accessible and transparent to the reader. Would you be willing to provide a PDF file per figure that

contains the original, uncropped and unprocessed scans of all or at least the key gels used in the manuscript and/or source data sets for relevant graphs? The files should be labeled with the appropriate figure/panel number, and in the case of gels, should have molecular weight markers; further annotation may be useful but is not essential. The files will be published online with the article as supplementary "Source Data" files. If you have any questions regarding this just contact me.

7) Every published paper includes a 'Synopsis' to further enhance discoverability. Synopses are displayed on the journal webpage and are freely accessible to all readers. They include a short standfirst as well as 2-5 one sentence bullet points that summarise the paper. Please provide the synopsis including the short list of bullet points that summarise the key NEW findings. The bullet points should be designed to be complementary to the abstract - i.e. not repeat the same text. We encourage inclusion of key acronyms and quantitative information. Please use the passive voice. Please attach this information in a separate file or send them by email, we will incorporate it accordingly. You are also welcome to suggest a striking image or visual abstract to illustrate your article. If you do please provide a jpeg file 550 px-wide x 400-px high.

8) We now mandate that ALL corresponding authors list an ORCID digital identifier. You may acquire one through our web platform upon submission and the procedure takes <90 seconds to complete. We also encourage co-authors to supply an ORCID identifier, which will be linked to their name for unambiguous name identification.

Please submit your revised manuscript within two weeks.

I look forward to reading a new revised version of your manuscript as soon as possible.

***** Reviewer's comments *****

Referee #1 (Comments on Novelty/Model System):

This is a revised manuscript in which the authors have addressed appropriately all comments of the reviewers.

Referee #1 (Remarks):

First, the author's concept of neurodegenerative disorders is much! too narrow. Neurodegeneration is defined as synaptic and neuronal loss and clearly not only caused by neurotoxic peptides since multy infarct dementia is a major cause, causing mixed AD/vaD which account for about 40-50% of all dementia cases (50% of AD cases carry the ApoE4 allele!). Please amend. Second, what is decreased in the CSF of AD patients is A β 42 and not total A β 40! Please amend! Fourth, what appears to be established for sporadic AD is a A β 42 clearance probleme (see Bateman et al.). Please amend the part addressed in response to Comment 6 of reviewer one. "Aggregation-associated" is not proven as cause!

Referee #2 (Comments on Novelty/Model System):

There are some aspects of the ELISA data that I still do not understand. The data on "soluble" Abeta are inconsistent - 10 fold higher in "older" animals. I am not sure how to rate the medical impact because I still do not fully grasp what the therapeutic dose was.

Referee #2 (Remarks):

The manuscript is much improved, but there are still some issues that remain. I do not fully understand the ELISA data, particularly what is defined as soluble Abeta. What is the Abeta soluble in? The values for soluble Abeta are 10 fold higher in mice aged longer, which indicates to me that the "soluble" fraction is actually also capturing some of the aggregated Abeta. The methods indicates GdnHCl was used in the buffer- which could disaggregate some of

the diffusely deposited Abeta. It might be less confusing to just focus on the formic acid insoluble Abeta.

The authors still do not explicitly state what they think the steady state level of the peptide is in the brains of the mice that showed behavioral and PET image improvements. The lack of change in plaque numbers suggests that most of the Abeta reduction is diffusely deposited Abeta. It might be best to give the dose in mg/kg (standard drug language).

The Results section contains language that is really more appropriate for Discussion.

2nd Revision - authors' response

24 January 2017

RESPONSE TO COMMENTS OF REVIEWERS

We thank the reviewers for their appreciation of the value of this study and their insightful comments. In response, we have revised our manuscript to address the issues that are raised in their 2nd reviews.

Reviewer 1:

Comment 1: First, the author's concept of neurodegenerative disorders is much! too narrow. Neurodegeneration is defined as synaptic and neuronal loss and clearly not only caused by neurotoxic peptides since multy infarct dementia is a major cause, causing mixed AD/vaD which account for about 40-50% of all dementia cases (50% of AD cases carry the ApoE4 allele!). Please amend.

Our reply:

We agree with the reviewer that neurodegeneration is a multifactorial process, including vascular pathology as the reviewer points out. Recently, misfolded proteinaceous aggregates have been shown to possess prion-like property that are instrumental to neurodegeneration in these neurodegenerative diseases including AD, PD and ALS etc. by several elegant studies. Since our study targets the misfolded aggregates, we did not go into details on other important factors as reviewer has mentioned. To make this point clear we have modified the texts:

*“Neurodegenerative diseases encompass a heterogeneous group of neurological diseases characterized by synaptic and neuronal losses caused by multiple factors. Misfolded proteinaceous aggregates which exist in a variety of these diseases besides AD, including Parkinson’s disease, Huntington’s disease, amyotrophic lateral sclerosis are considered one of them, and may **cause or contribute to** these diseases through their prion-like property (Kim and Holtzman Science 2010. 330 (6006): 918-919; de Calignon et al., Neuron. 73:685–697; Luk et al. Science. 2012 Nov 16; 338(6109): 949–953.; Smethurst et al. NBD 2016. 96:236-247).”* (page 6; lines 1-7)

Comment 2: Second, what is decreased in the CSF of AD patients is Aβ42 and not total Aβ40! Please amend! Fourth, what appears to be established for sporadic AD is a Aβ42 clearance problem (see Bateman et al.). Please amend the part addressed in response to Comment 6 of reviewer one. "Aggregation-associated" is not proven as cause!

Our reply:

We thank reviewer for pointing this out. Indeed, the decrease in Aβ42 is a specific CSF biomarker for AD. In response, we have changed the corresponding text as *“One important biomarker in AD diagnosis is the decrease of Aβ₄₂ level in the cerebral spinal fluid due to Aβ aggregation”* (page 13, line 10).

Concerning the statement that aggregation as the cause or not, please see the response to comment #1. Briefly, since the prion-like property of the protein aggregates has been clearly shown, it seems safe to argue they are at least partially responsible for pathogenesis of these diseases. So as above stated, we have revised the text to *“cause or contribute to”* to avoid potential confusions. (page 6, line 5)

Reviewer 2:

Comment 1: The manuscript is much improved, but there are still some issues that remain. I do not fully understand the ELISA data, particularly what is defined as soluble Abeta. What is the Abeta soluble in? The values for soluble Abeta are 10 fold higher in mice aged longer, which indicates to me that the "soluble" fraction is actually also capturing some of the

aggregated Abeta. The methods indicates GdnHCl was used in the buffer- which could disaggregate some of the diffusely deposited Abeta. It might less confusing to just focus on the formic acid insoluble Abeta.

Our reply:

We thank reviewer for pointing out this potential confusion. In fact, there were 2 different protocols used for ELISA assays. For ELISA assays of 8 month-old mice, since the amyloid plaques had not yet developed, A β was largely soluble in extraction buffer containing GdnHCl as instructed by the manufacturer. So there was not yet a need to separate the A β into soluble or insoluble pools at this time. However, for ELISA assays of older (13-month) mice, amyloid plaques had formed and the A β in plaques may not be entirely extractable with GdnHCl, so we first used the extraction buffer with SDS to extract SDS-soluble fraction, and then with formic acid to extract A β from the SDS-insoluble amyloid plaques according to van Groen *et al* (2013). We have revised the manuscript:

“The total levels of A β_{40} and A β_{42} in the brain homogenate of 8-month-old mice were detected using ELISA kits (Invitrogen, MD, USA) according to the manufacturer’s instructions. Briefly, the cortical or hippocampal tissue was homogenized in ice-cold cell extraction buffer provided in the kit with protease inhibitor cocktail (Sigma, St. Louis, USA) and centrifuged at 15,000 g at 4°C for 10 minutes. In the protocol, 5 M GdnHCl was used in A β extraction buffer.

*Because senile plaques started to form in mice older than 8 months of age, A β in these plaques might not be extracted by SDS- or GdnHCl (Kawarabayashi *et al*, 2001). A β was separated into the SDS-soluble and SDS-insoluble fractions. Formic acid (FA) was used instead to extract SDS-insoluble A β_{40} and A β_{42} in plaques from the brain homogenate of the 13-month-old mice (van Groen *et al*, 2013). The brain tissues were first homogenized in the A β extraction buffer containing 20mM Tris-HCl (pH 7.6), 137mM NaCl, 1% Triton X-100, 2% SDS, and protease inhibitor cocktail and centrifuged at 20,000 g for 20 minutes at 4°C. The supernatant was the SDS-soluble fraction for ELISA measurement of soluble A β . The pellet was then dissolved in 70% FA, sonicated for 1 minute, and then centrifuged at 20,000 g for 20 minutes at 4°C. The resultant supernatant was the SDS-insoluble fraction. This fraction was neutralized with 20 volumes of 1 M Tris base before ELISA measurement. Total protein concentrations of the SDS-soluble and SDS-insoluble fractions were quantified using the Bradford protein assay (Bio-Rad, Hercules, CA, USA). A β amount of each fraction was normalized to the total protein concentration of that fraction for comparison.”* (page 25, line 1- page 26, line 3).

The amount of A β was normalized by total protein in its own fraction. However, to compare the amount of A β peptide in soluble and insoluble fractions needs to take the volume of the extraction buffer into consideration. However, in our analyses, we did not adjust the final amount of A β in each fraction by the factor of buffer volume. So the numbers of A β in soluble and insoluble fractions in Fig.4 C & 4D cannot be directly compared. Thus, the values for soluble A β are not 10 fold higher in mice aged longer. We apologize for this confusion.

Comment 2: The authors still do not explicitly state what they think the steady state level of the peptide is in the brains of the mice that showed behavioral and PET image improvements. The lack of change in plaque numbers suggests that most of the Abeta reduction is diffusely deposited Abeta. It might be best to give the dose in mg/kg (standard drug language).

Our reply:

Indeed, we did not investigate the steady state level of our therapeutic peptide in the brain. In response to the comment, we conducted one new set of animal experiment to measure the peptide amount in the brains of the mice treated with FITC-labeled peptide at the therapeutic dosage that resulted in positive responses in behavior and PET imaging studies. Specifically, the mice received the FITC-labelled peptide (2 nmole/mouse/day) twice, followed by a 24-hour break, then treated again. We measured the FITC-labelled peptide concentration in the brains of mice every 6 h or 12 h. However, the fluorescence signals were low, and below 0.2 nmole (revised Fig. S6). Thus the level was close to or below the margin that could be reliably detected by our method. As stated in our discussion, the dose of our peptide used in this study is in fact quite low compared with previous studies. Thus, these results are consistent with our other experiments which showed a high efficacy of our peptide in blocking A β from forming fibrils at a ratio of 1:10 (Fig. 1) and therefore its toxicity. Since a higher dose we used in experiments in order to detect intracerebral peptide (Fig. 5) resulted in a higher level of intracerebral peptide, it is possible that the therapeutic effect of our peptide may be further enhanced by higher dosages. The results are shown as new supplemental Fig. S6. The previous Fig. S6 and Fig. S7 were renamed as Fig. S7 and Fig. S8, respectively, in this

revised version. We also included the description of this experiment in the method section on page 24 and revised the Fig. S3 to include this set of the designs of animal experiments. We have revised the manuscript:

“In this study, we have demonstrated the feasibility of intranasal administration of therapeutic peptidic prodrugs. When combined with technology in delivery, our study showed a proof of therapeutic principle for neurodegenerative diseases through intranasal delivery. The dose used in this study was only 2 nanomoles (6 mg) per day, which was quite low compared with previous studies (Frydman-Marom et al, 2009; Funke et al, 2010; Permanne et al, 2002; van Groen et al, 2008). Using this dosage we attempted to investigate the level of the therapeutic peptide in the brain during consecutive intranasal treatment (experimental set 5 in Appendix Fig. S3 and Fig. S6). However, the peptide concentration was low, and could not be reliably detected. As shown in Fig. 5, after three consecutive treatments at higher amount (9 nmoles), there was 5.16 nmole of the peptide in the brain at 6 hours after the final treatment, and 3.62 nmole of the peptide in the brain 24 hours after the final treatment. Although the current method was geared toward maximizing our ability to detect the intracerebral peptide rather than producing an accurate number in its efficiency in brain entrance, an estimated value was still achievable. Since the treatment continued for 3 days, the amount of intracerebral peptide before the 3rd dose was expected not to be more than 3.62 nmole observed 24 hours after the 3rd treatment. Thus, at least 1.54 nmole (5.16 minus 3.62) or 17% of the daily dose of 9 nmole peptide entered brain. These results indicate that this peptide had a reasonably high therapeutic efficacy. Future studies will be conducted for optimal dosage.” (from page 17, line 2 to page 18, line 3)

Concerning the plaque number, it is certainly possible as the reviewer suggests. Our peptide reduced the size of individual amyloid plaques and also the total areas of amyloid plaques, indicating the effect on reducing the growth of these plaques. So alternatively, when we started treatment, the cores of plaques might have already formed at 4 months, but our peptide slowed down the speed of the accumulation of the transgenic A β . This was also included in the 4th paragraph of discussion in the revised manuscript. *“The peptide treatment did not significantly decrease the numbers of the ThS-positive amyloid plaques, but reduced the size of the individual plaques and the total area of these plaques. One possibility is that most of the the A β reduction is diffusely deposited A β . Alternatively, when we started treatment, the cores of plaques might have already formed at 4 months, but our peptide slowed down the speed of the accumulation of the transgenic A β of these plaques. Moreover, when we quantified SDS-soluble A β and SDS-insoluble A β separately (Fig.4C-F), we found that SDS-insoluble A β reduced after peptide treatment whereas SDS-soluble A β increased. A β accumulation is due to the imbalance of A β production and A β degradation. Our peptide treatment likely functions to inhibit A β from self-association, but may not directly impact on the A β degradation rate. The clearance of excessive A β depends on several Ab-degrading enzymes, such as neprilysin (the most important one) and insulin-degrading enzyme, which were found to be downregulated in old mice (Caccamo et al, 2005). However, by preventing A β from aggregation, our peptide could render it more accessible to these Ab-degrading enzymes and/or other degradation machinery in the brain. Recently, it has been reported that polyhydroxycurcuminoids upregulate neprilysin in the brain (Chen et al, 2016). Combining the peptide inhibitor and the neprilysin activator might additively enhance Ab clearance.”(page 18, line 4 – page 19, line 4)*

We thank the reviewer for bringing up the issue of body weight. The bodyweight of mice increased from ~25 g to 30-35 g with age (from 4 months to 13 months). However, we used the fixed amount of therapeutic peptide throughout our studies, and did not increase the dose according to the change of body weight. To make it more clear, we described the dosages as nmole/mouse/day (See pages 13-14 and 23-24 and the revised Fig.S3). Since the body weight continued to rise, this could have offset to some extent the therapeutic effect of the peptide in our final analyses.

Comment 3: The Results section contains language that is really more appropriate for Discussion.

Our reply:

We thank the reviewer for pointing this out. We agree. Since the Reviewer did not enlist specific parts of the manuscript that needs to be changed, three paragraphs are decided to be more appropriate for the discussion session, and moved.

- (1) *“The clearance of excessive A β depends on several A β -degrading enzymes, such as neprilysin (the most important one) and insulin-degrading enzyme, which were found to*

be downregulated in old mice (Caccamo et al, 2005). By preventing A β from aggregation, our peptide could render it more accessible to these A β -degrading enzymes and/or other degradation machinery in the brain. Recently, it has been reported that polyhydroxycurcuminoids upregulate neprilysin in the brain (Chen et al, 2016). Combining the peptide inhibitor and the neprilysin activator might additively enhance A β clearance.” on original page 13. We have moved them to the 4th paragraph of discussion in the revised manuscript (pages 18, line 14-page 19 line 4)

- (2) *“There was still 3.62 nmole of the peptide in the brain 24 hours after treatment. Since the peptide was given at an interval of 24 hours, the amount of intracerebral peptide should be 3.62 nmole or more throughout the therapeutic period. The current method was geared toward maximizing our ability to detect the intracerebral peptide, rather than to produce an accurate number in its efficiency in brain entrance. However, an estimated value was still achievable. Since the treatment continued for 3 days, the amount of intracerebral peptide before the 3rd dose was expected not to be more than 3.62 nmole observed 24 hours after the 3rd treatment. Thus, at least 1.54 nmole (5.16 minus 3.62) or 17% of the daily dose of 9 nmole peptide entered brain.” on original page 15. We have moved them to the 3rd paragraph of discussion in the revised manuscript with modifications. (pages 17, line 8 - page18, line 3)*

“Using this dosage we attempted to investigate the level of the therapeutic peptide in the brain during consecutive intranasal treatment (experimental set 5 in Appendix Fig. S3 and Fig. S6). However, the peptide concentration was low, and could not be reliably detected. As shown in Fig. 5, after three consecutive treatments at higher amount (9 nmoles), there was 5.16 nmole of the peptide in the brain at 6 hours after the final treatment, and 3.62 nmole of the peptide in the brain 24 hours after the final treatment. Although the current method was geared toward maximizing our ability to detect the intracerebral peptide rather than producing an accurate number in its efficiency in brain entrance, an estimated value was still achievable. Since the treatment continued for 3 days, the amount of intracerebral peptide before the 3rd dose was expected not to be more than 3.62 nmole observed 24 hours after the 3rd treatment. Thus, at least 1.54 nmole (5.16 minus 3.62) or 17% of the daily dose of 9 nmole peptide entered brain. These results indicate that this peptide had a reasonably high therapeutic efficacy. Future studies will be conducted for optimal dosage.”

- (3) *“To determine if the peptide treatment induced an antibody response against A β peptide, the serum of the mice treated for 15 days were tested, and showed no evidence of immunoreactivity against the peptide (experimental set 5 in Appendix Fig. S3 and Fig. S8).” on original page 16 was moved to the 6th paragraph in the discussion section on the revised page19, lines 11-14.*

3rd Editorial Decision

27 February 2017

Thank you for the submission of your revised manuscript to EMBO Molecular Medicine. We are sorry that it has taken so long to get back to you on your manuscript. In this case we experienced some difficulties in obtaining an evaluation from the reviewer. Finally, we also needed to discuss the evaluation further as you will see below, which required additional time.

As you will see reviewer 2 is quite reserved and very critical and raises new doubts (which potentially are caveats and thus justified). These include a serious concern about the mouse strain, which s/he suggests might be blind. The other two issues are persisting concerns over the ELISAs and lack of brain peptide measurements.

Given the contrasting opinions, I exceptionally asked reviewer 1 to provide additional cross-commenting. Reviewer 1 replied by confirming that s/he is not concerned about the solubility and brain peptide issues and finds figures 3 and 4 to be consistent. As for the mice, s/he does not agree that the visual cue tasks appear to those of blind mice.

Based on reviewer 1's further comments and after internal discussion, I am prepared to move

forward on your manuscript. I will evaluate your next, final version of your manuscript at the editorial level, provided the items are fully addressed. Specifically, while I will not be asking you to provide further experimentation (unless you have additional data at hand) please address the concerns raised by reviewer 2 in a point-by-point rebuttal. I find the doubts on the mouse strain of special importance and therefore I must ask you to provide a definitive answer, if necessary by providing genotyping data.

Please submit your revised manuscript within two weeks. I look forward to seeing a revised form of your manuscript as soon as possible.

***** Reviewer's comments *****

Referee #2 (Comments on Novelty/Model System):

1) The use of 2 different ELISA protocols un-necessarily creates confusion for the reader. The protocol used in Figure 4 could have also been used effectively in Figure 3. I am not sure that all of the Abeta in 8 month old APP^{swe}/PS1^{dE9} mice would be completely soluble in the GdnHCl buffer. It is interesting that what is being measured is reduced in these mice by peptide treatment, but I am not entirely sure that the authors the method used really measures all of the Abeta that may be present in the brains of these mice.

2) The approach is novel.

3) There could be medical impact but I am unconvinced by the data presented here.

4) I have now noticed an issue with the behavior study that I had over-looked before. I had assumed that the authors were using the B6 congenic background for behavioral studies, but I now see that the authors state that they use APP^{swe}/PS1^{dE9} mice in the B6C3 hybrid background. The authors do not acknowledge that the C3H strain component of this hybrid carries the retinal degeneration gene (rd). There is no mention of genotyping for the rd mutation in generating the cohorts of mice. Crosses of male transgenic APP^{swe}/PS1^{dE9} mice to female B6C3F1 breeders may produce mice that are homozygous for the rd mutation, and blind. This can be avoided by genotyping the male breeders for the rd mutation to find mice that are homozygous WT at the rd locus. Mice that are homozygous for the rd mutation mice should not be used in visual cue tasks like the water maze. The authors do not explicitly indicate whether the control B6C3 mice for the behavioral work were littermate controls. The authors do not explicitly indicate whether litters were randomized for treatment and vehicle.

Referee #2 (Remarks):

The study described by Cheng et al still has some very serious flaws that lead me to recommend against publication. There are multiple aspects of the data that cause concern.

1) The use of 2 different ELISA protocols un-necessarily creates confusion for the reader. The protocol used in Figure 4 could have also been used effectively in Figure 3. I am not sure that all of the Abeta in 8 month old APP^{swe}/PS1^{dE9} mice would be completely soluble in the GdnHCl buffer. It is interesting that what is being measured is reduced in these mice by peptide treatment, but I am not entirely sure that the authors the method used really measures all of the Abeta that may be present in the brains of these mice.

2) I have also now noticed an issue with the behavior study that I had over-looked before. I had assumed that the authors were using the B6 congenic background for behavioral studies, but I now see that the authors state that they use APP^{swe}/PS1^{dE9} mice in the B6C3 hybrid background. The authors do not acknowledge that the C3H strain component of this hybrid carries the retinal degeneration gene (rd). There is no mention of genotyping for the rd mutation in generating the cohorts of mice. Crosses of male transgenic APP^{swe}/PS1^{dE9} mice to female B6C3F1 breeders will produce mice that are homozygous for the rd mutation, and blind. This can be avoided by genotyping the male breeders for the rd mutation to find mice that are homozygous WT at the rd locus. Mice that are homozygous for the rd mutation mice should not be used in visual cue tasks like the water maze. The authors do not indicate that they assess visual acuity of the mice (visible

platform trials). The authors do not explicitly indicate whether the control B6C3 mice for the behavioral work were littermate controls. The authors do not explicitly indicate whether litters were randomized for treatment and vehicle.

3) In Figure 4, the location of the more intense PiB uptake is not entirely what I would expect for the APPswe/PS1dE9 model. There is some cortical/hippocampal signal, but most of the signal is located in the striatum and ventral midbrain. These regions generally have fewer deposits than the cortex or hippocampus. The number of plaques in the cortex and hippocampus does not change by treatment.

4) The authors have not been able to determine the therapeutic concentration of the peptide in brain. The data suggest that the effects observed occur at sub-nanomolar concentrations of the peptide.

3rd Revision - authors' response

01 March 2017

RESPONSE TO COMMENTS OF REVIEWER 2

Comment 1: The use of 2 different ELISA protocols un-necessarily creates confusion for the reader. The protocol used in Figure 4 could have also been used effectively in Figure 3. I am not sure that all of the Abeta in 8 month old APPswe/PS1dE9 mice would be completely soluble in the GdnHCl buffer. It is interesting that what is being measured is reduced in these mice by peptide treatment, but I am not entirely sure that the authors' method used really measures all of the Abeta that may be present in the brains of these mice.

Our reply:

We agree with the reviewer that two protocols indeed might create confusion for the readers. A β extractions for Fig. 3 and Fig. 4 were not done at the same time. We did the A β extraction for the 8-month-old APPswe/PS1dE9 mice earlier and we exactly followed the extraction procedures in the ELISA kit. We can not guarantee whether A β peptides in the 8-month-old APPswe/PS1dE9 mice are completely soluble in the GdnHCl buffer. But the extractions for the peptide-treated or untreated APPswe/PS1dE9 mice were done at the same time, by the same person, and using the same protocol. We can compare the A β levels under the same extraction condition.

Comment 2: I have now noticed an issue with the behavior study that I had over-looked before. I had assumed that the authors were using the B6 congenic background for behavioral studies, but I now see that the authors state that they use APPswe/PS1dE9 mice in the B6C3 hybrid background. The authors do not acknowledge that the C3H strain component of this hybrid carries the retinal degeneration gene (rd). There is no mention of genotyping for the rd mutation in generating the cohorts of mice. Crosses of male transgenic APPswe/PS1dE9 mice to female B6C3F1 breeders may produce mice that are homozygous for the rd mutation, and blind. This can be avoided by genotyping the male breeders for the rd mutation to find mice that are homozygous WT at the rd locus. Mice that are homozygous for the rd mutation should not be used in visual cue tasks like the water maze. The authors do not explicitly indicate whether the control B6C3 mice for the behavioral work were littermate controls. The authors do not explicitly indicate whether litters were randomized for treatment and vehicle.

Our reply: I am not an animal expert so I call our national animal center to confirm about the animal strain we used in breeding and the other experiments. Actually we use **male transgenic APPswe/PS1dE9 mice and female C57BL/6JNarl mice** to do the breeding. C57BL/6JNarl mice is also called B6, but not B6C3. The typo was our mistake. I have corrected the mistake in the revised manuscript.

By the way, I have enquired Jackson Lab about this issue. Here is their reply:

Dear Dr. Rita Chen,

Thank you for your inquiry.

We have 2 APPswe/PS1dE9 mouse strains:

B6C3-Tg(APPswe,PS1dE9)85Dbo/Mmjax (Stock# 004462)

www.jax.org/strain/004462

and

B6.Cg-Tg(APPswe,PS1dE9)85Dbo/Mmjax (Stock# 005864)

<https://www.jax.org/strain/005864>

Strain 004462 is a closed colony and does not carry the retinal degeneration allele Pde6b<rd1>. It is mated noncarrier x Hemizygote. We do not use strain B6C3F1 #100010 to maintain this strain.

Strain 005864 is congenic to the C57BL/6J genetic background and likewise also does not carry the retinal degeneration allele Pde6b<rd1>.

If you were to cross in strain B6C3F1 (Stock# 100010, <https://www.jax.org/strain/100010>) to either of the above transgenic mice, you would be introducing the Pde6b<rd1> allele (contributed from the C3H strain) into your colony. If this were to become homozygous in any of your mice, then yes, the mice would exhibit retinal degeneration.

Let me know if you have other questions.

Please indicate your level of satisfaction with the handling of this inquiry by answering a few questions at the following link: <https://www.surveymonkey.com/s/JAXtechsupport>

Sincerely,

Janine

Janine Low-Marchelli, Ph.D.

Technical Information Scientist

The Jackson Laboratory

Ph: 1-800-422-MICE (6423) or 207-288-5845 (International)

Email: micotech@jax.org

.....

Therefore, since we did not use C3 strain in the breeding and B6C3-Tg(APP^{swe},PSEN1^{dE9})85Dbo/Mmjax does not carry the retinal degeneration allele Pde6b<rd1>. No mice will carry the retinal degeneration gene.

Corresponding Author Name: Rita P-Y Chen

Journal Submitted to: EMBO Mol Med

Manuscript Number: EMM-2016-06666-T